# Crystal structure of undecaprenyl-pyrophosphate phosphatase and its role in peptidoglycan biosynthesis

Meriem El Ghachi[1], Nicole Howe[2], Chia-Ying Huang[2,3], Vincent Olieric[3], Rangana Warshamanage[3], Thierry Touzé[4], Dietmar Weichert [2], Phillip J. Stansfeld[5], Meitian Wang[3], Fred Kerff[1] & Martin Caffrey[2]

As a protective envelope surrounding the bacterial cell, the peptidoglycan sacculus is a site of vulnerability and an antibiotic target. Peptidoglycan components, assembled in the cytoplasm, are shuttled across the membrane in a cycle that uses undecaprenyl-phosphate. A product of peptidoglycan synthesis, undecaprenyl-pyrophosphate, is converted to undecaprenyl-phosphate for reuse in the cycle by the membrane integral pyrophosphatase, BacA. To understand how BacA functions, we determine its crystal structure at 2.6 Å resolution. The enzyme is open to the periplasm and to the periplasmic leaflet via a pocket that extends into the membrane. Conserved residues map to the pocket where pyrophosphorolysis occurs. BacA incorporates an interdigitated inverted topology repeat, a topology type thus far only reported in transporters and channels. This unique topology raises issues regarding the ancestry of BacA, the possibility that BacA has alternate active sites on either side of the membrane and its possible function as a flippase.

[1] Centre d'Ingénierie des Protéines, InBioS, Université de Liège, allée du 6 Août 19, Bât B5a, 4000 Liège, Belgium. [2] Membrane Structural and Functional Biology (MS&FB) Group, School of Medicine and School of Biochemistry and Immunology, Trinity College Dublin, Dublin 2 D02 R590, Ireland. [3] Swiss Light Source, Paul Scherrer Institute, CH-5232 Villigen, Switzerland. [4] Institute for Integrative Biology of the Cell (I2BC), CEA, CNRS, Université Paris-Sud, Université Paris-Saclay, 91190 Gif-sur-Yvette, Cedex, France. [5] Department of Biochemistry, University of Oxford, South Parks Road, Oxford OX1 3QU, UK. These authors contributed equally: Meriem El Ghachi, Nicole Howe. Correspondence and requests for materials should be addressed to M.C. (email: martin.caffrey@tcd.ie)

Bacteria have a protective cell wall composed of lipid, protein and carbohydrate that is essential for growth, pathogenicity and virulence. The carbohydrate component exists as a complex in the form of peptidoglycan, lipopolysaccharide, and teichoic acid. As an exoskeletal structure, peptidoglycan envelops the cell just outside the plasmamembrane defining cell shape and size and providing protection against osmotic lysis[1]. Cell wall components and the enzymes involved in their synthesis are important antibiotic targets[2,3].

Peptidoglycan components are synthesized in the cytoplasm and are transferred across the membrane for glycan polymerization by means of a peptidoglycan component synthesis (PCS) cycle that spans the plasmamembrane (Fig. 1)[4,5]. Half of the cycle operates in the membrane's inner leaflet where sugar and sugar-peptides are linked together on a polyisoprene carrier molecule, undecaprenyl-phosphate (C55P)[5,6]. The mature form of the carrier-cargo molecule, lipid II, is transferred across the membrane by a flippase where it is anchored in the outer leaflet[7]. There, its disaccharide-peptide cargo is used in glycan polymerization and peptidoglycan cross-linking with undecaprenyl-pyrophosphate (C55PP) as the lipid product. For the cycle to begin again, C55PP in the outer leaflet must be converted to C55P by a pyrophosphatase enzyme. The C55P is then returned to the inner leaflet whereupon it can re-enter the PCS cycle.

Along with C55P, C55PP is also found in the inner leaflet of the cell membrane. It derives from de novo synthesis in a series of reactions catalyzed by the cytosolic enzyme undecaprenyl-pyrophosphate synthase, UppS (Fig. 1)[5,11]. To enter the PCS cycle as C55P, de novo synthesized C55PP must be acted on by a second pyrophosphatase. It would appear therefore that for the cycle to operate, at least two pyrophosphatases, one to the periplasmic side and another to the cytoplasmic side of the

membrane, are required. Because of their importance in cell envelope synthesis and their attractiveness as potential drug targets considerable effort has been devoted to identifying these pyrophosphatases[2,3]. In *Escherichia coli*, three quarters of the relevant pyrophosphatase activity in isolated membranes has been ascribed to BacA (also known as UppP)[6]. The remaining activity was attributed to three members of the phosphatidic acid phosphatase (PAP2) family that includes PgpB, YeiU/LpxT and YbjG[6]. All contain the PAP2 consensus sequence motif[12]. PgpB, the most extensively studied of the three, is involved in phosphatidylglycerol synthesis[13,14]. It also has pyrophosphatase activity and functions with C55PP as substrate[14]. The structure of *E. coli* PgpB is known and the enzyme has been shown to have its active site facing the periplasm[14–16]. It appears likely therefore that PgpB contributes to the regeneration of C55P from C55PP on the periplasmic side of the membrane.

BacA, which does not contain the PAP2 consensus sequence, functions as a pyrophosphatase with C55PP as a substrate. Its physiological importance was highlighted when it was demonstrated that an inactive BacA mutant in *Streptococcus pneumoniae* was 160,000-fold more sensitive to the C55PP-binding antibiotic, bacitracin. Further, a 40,000-fold reduction in bacterial load was detected for this BacA mutant in a mouse lung infection model[17]. Remarkably, BacA in *E. coli* shares the very high sequence identity of 63% with homologs in other Gram-negative species. Having stood the test of evolutionary time its sequence is highly effective with two thirds of its residues presumably playing key roles in folding, stability and/or function. However, such a high level of identity makes it difficult to pin-point catalytically relevant residues. Resorting to alignments of *E. coli* BacA with more divergent Gram-positive species, two regions (referred to hereafter as BacA1 and BacA2 sequence motifs; Supplementary

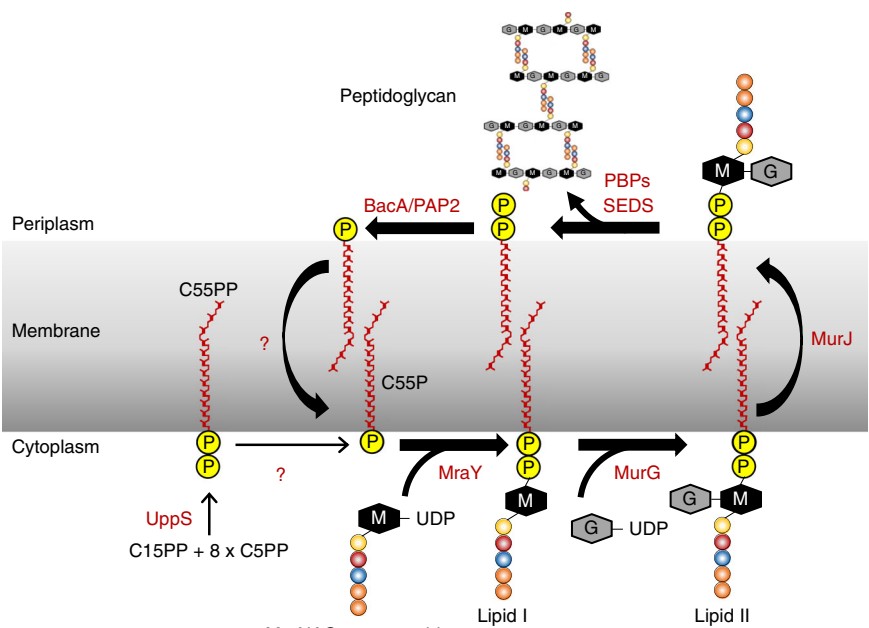

**Fig. 1** Peptidoglycan component synthesis (PCS) cycle. Undecaprenyl-pyrophosphate (C55PP) is synthesized by UppS to the cytoplasmic side of the inner membrane. It is dephosphorylated forming undecaprenyl-phosphate (C55P), a glycan subunit carrier, by a pyrophosphatase of unknown identity. The enzymes MraY and MurG catalyze the successive transfer of the MurNAc-pentapeptide and GlcNAc motifs from nucleotide precursors to the C55P carrier, generating lipid I and lipid II, respectively. Lipid II is translocated by a dedicated flippase, MurJ[8], to the periplasmic side of the membrane where polymerization of peptidoglycan catalyzed by the penicillin-binding proteins (PBPs) and the shape, elongation, division and sporulation (SEDS) family of proteins occur. The SEDS proteins, RodA[9] and FtsW[10] may also be involved in flipping lipid II. C55PP, released as a by-product of transglycosylation reactions, is dephosphorylated forming C55P, which is shuttled back by an unknown mechanism to the inner leaflet of the membrane for reuse in the PCS cycle

Fig. 1) stood out that retained a high level of identity. These have been implicated as being important in substrate binding and catalysis[18,19].

In an effort to decipher BacA's membrane topology and to identify the side of the membrane on which its active site resides, the protein has been subjected to a range of bioinformatics and biochemical studies (Fig. 2). To address many of the disparities raised by these investigations and to shed light on the mechanism of action of this physiologically and therapeutically important membrane enzyme, a crystal structure of BacA was called for.

We report the crystal structure of BacA at a resolution of 2.6 Å. The enzyme is open to the periplasm and to the periplasmic leaflet via a pocket that extends into the membrane. Conserved residues map to the pocket where pyrophosphorolysis takes place. Remarkably, BacA incorporates an interdigitated inverted topology repeat. This topology type has thus far only been reported in transporters and channels. As an unexpected topology for an enzyme, it raises issues concerning BacA's ancestry in gene duplication and its possible function as a flippase.

## Results

Crystallization trials of BacA from *E. coli* by the in meso (lipid cubic phase, LCP) method[20] provided first crystals at 20 °C with mono-olein (9 MAG) as host lipid. Optimization of precipitant conditions gave diffraction to a resolution of 2.8 Å using harvested, cryo-cooled micro-crystals. In an effort to further optimize diffraction and to avoid the cumbersome and potential damaging effects of crystal harvesting, a new in meso in situ X-ray crystallography (IMISX) method was implemented[21]. The IMISX method enables high-throughput serial diffraction measurements to be made on micro-crystals where and as they grow without recourse to direct crystal harvesting. Data were collected at cryo-temperatures (IMISXcryo) to extend the lifetime of the crystal in the X-ray beam. Consistently, the IMISXcryo crystals provided better quality data to higher resolution in comparison with traditional, loop harvested crystals. A structure to 2.6 Å resolution was finally obtained by mercury-SAD phasing using 53 crystals grown with protein pre-treated with mercury chloride (Table 1, Supplementary Fig. 2). The asymmetric unit consists of a single protein molecule, two mercury atoms, a Tris molecule and two lipids identified as monoolein. Type I crystal packing, a characteristic of in meso grown crystals, was observed with layers of protein reminiscent of packing in a membrane (Supplementary Fig. 3).

**Table 1 Data collection and refinement statistics for BacA-Hg**

| PDB ID | 5OON |
| --- | --- |
| *Data collection* | |
| Space group | C222 |
| *Cell dimensions* | |
| *a, b, c* (Å) | 113.26, 145.00, 40.49 |
| *α, β, γ* (°) | 90, 90, 90 |
| Beamline | X06SA-PXI |
| Wavelength (Å) | 1.9 |
| No. of wells | 2 |
| No. of crystals | 53 |
| Total data (°) | 1030 |
| Resolution (Å) | 44.45–2.60 (2.67–2.60) |
| $R_{meas}$ | 0.40 (2.03) |
| $I/σ_I$ | 8.75 (1.87) |
| Completeness (%) | 100 (99.90) |
| Multiplicity | 17.91 (12.33) |
| $CC_{1/2}$ | 0.99 (0.25) |
| $CC_{anom}$ | 0.24 |
| Mosaicity (°) | 0.142 |
| Phasing | Hg-SAD |
| Resolution range (Å) | 44.45–2.80 |
| Heavy atoms sites | 2 Hg |
| Correlation coefficient (all/weak) | 49.3/27.6 |
| *Refinement* | |
| Resolution (Å) | 44.45–2.60 |
| No. of reflections | 19,853/990 |
| $R_{work}/R_{free}$ | 0.21/0.24 |
| *R.m.s. deviations* | |
| Bond length (Å) | 0.006 |
| Bond angle (°) | 0.865 |
| B-factor (Å$^2$) | |
| Protein | 50.85 |
| Hg | 79.89 |
| 9.9 MAG and Tris | 56.15 |
| Water | 49.98 |
| *Ramachandran Plot* | |
| Favored (%) | 98.51 |
| Allowed (%) | 1.49 |
| Outliers (%) | 0 |
| *MolProbity* | |
| Clash score | 6.23 |

Data processing statistics are reported with Friedel pairs separated. Values in parentheses are for the highest resolution shell

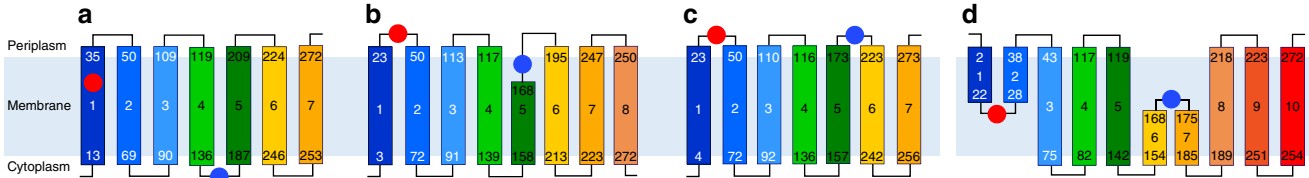

**Fig. 2** Proposed topology models. **a** The earliest proposed topology was based on a bioinformatics investigation, which concluded that the enzyme had seven membrane helices, a cytoplasmic N-terminus and an active site facing the cytoplasm[61]. The putative active site included the BacA2 sequence motif while the conserved BacA1 region mapped to the first membrane helix and was proposed to be involved in catalysis and in substrate specificity. **b** The second proposed model relied on bioinformatics, modeling, and functional and site-directed mutation work employing constructs of BacA fused with bacteriorhodopsin and green fluorescent protein[19]. This led to the conclusion that the protein had eight membrane helices with both N- and C-termini in the cytoplasm and an outward facing active site. **c** The third model involved an experimental investigation focussed on biochemistry with functional and mutational studies employing hybrids prepared from C-terminal truncations of BacA fused to β-lactamase[18]. This facilitated in vivo screening based on ampicillin resistance or susceptibility depending on which side of the membrane that β-lactamase resided. The model to emerge from this study had seven membrane helices with the N-terminus in the cytoplasm and the active site facing the periplasm. **d** The latest model was based on residue-residue co-evolution constraints applied in the Rosetta structure prediction program to generate a structure of BacA[24]. The modeled protein includes a pair of inverted helices and has twofold pseudosymmetry as observed in the structure determined experimentally in the current study. The membrane orientation in **d** was not defined. Putative catalytic residues Ser27 and R174 are indicated by red and blue dots, respectively

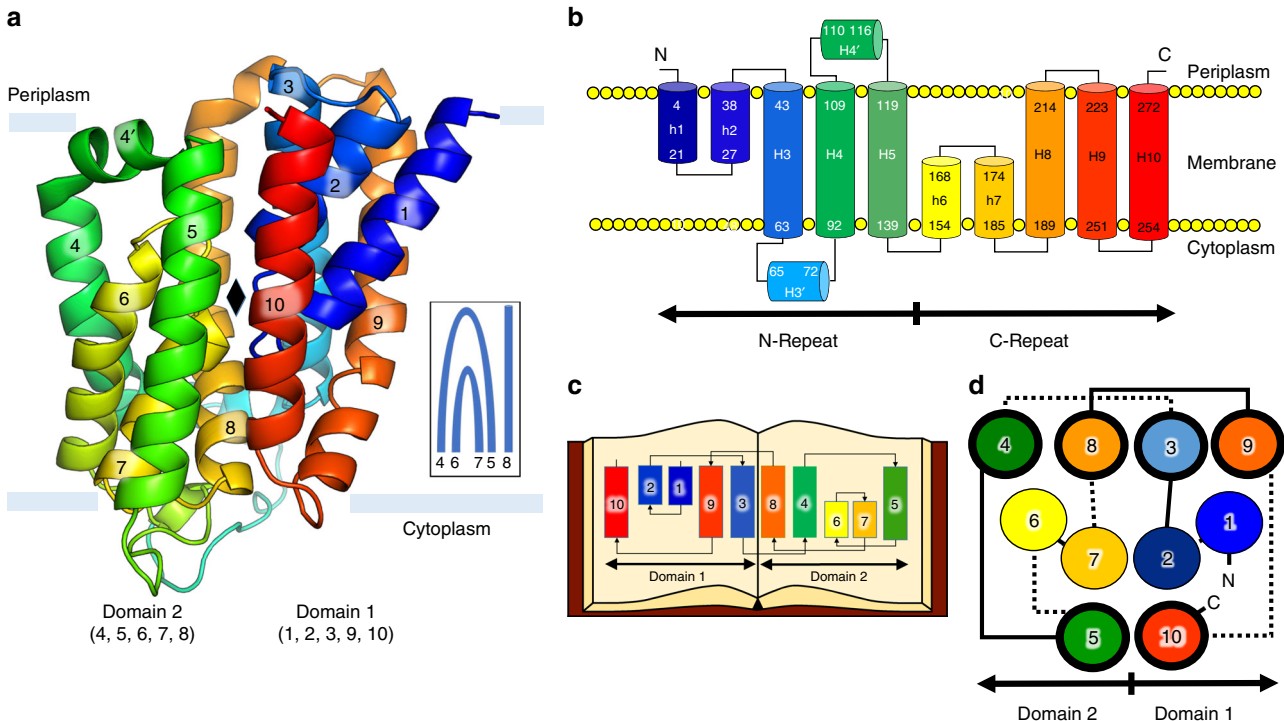

**Fig. 3** Crystal structure and topology of BacA from *E. coli*. **a** View from the membrane plane. The protein has two domains related by a twofold axis in the plane of the membrane. Each domain consists of a double arch motif and a buttress helix. The double arch includes an outer arch formed by two transmembrane helices and an inner arch in the form of a re-entrant helical pair. A schematized version of domain 2 is included in the inset. The active site resides toward the mid-plane of the membrane at the core of the protein where the domains and the re-entrant helix loops approach one another. The structure is shown in cartoon representation and rainbow color coded from N-terminus (blue) to C-terminus (red). Approximate location of the membrane boundaries are shown in gray. The $C_2$ axis of pseudosymmetry is marked with a diamond. **b** Topology of BacA in a simplified representation to highlight the N-repeat and C-repeat as elements of symmetry in the protein. **c** Bringing together the two domains can be visualized by laying the folded topology on an open book. Closing the book brings the elements in the two domains into alignment generating the fully folded protein. **d** Schematic representation of the secondary structure elements in the BacA structure viewed from the periplasm. Circles and lines correspond to helices and loops, respectively. Solid lines are in or face the periplasm. Dashed lines are in or face the cytoplasm. The view shows how helices in the N-repeat (helices 1–5) and C-repeat (helices 6–10) interdigitate and combine to create domains with helices from both repeats. Color coding follows that in **a**

**Overall architecture**. The protein consists of a collection of ten tightly bundled helices; six of these are transmembrane helices (H3-5, H8-10) (Fig. 3, Fig. 4). The remaining four exist as pairs of short, antiparallel re-entrant helices (h1 and h2, h6 and h7). Individual re-entrant helices are connected by short loops (L1–2 and L6–7, respectively) located in close proximity to one another toward the mid-section of the protein. With an equal number of transmembrane helices, both the N- and C-termini of BacA are on the same side of the membrane. Assuming the positive-inside rule[22] applies, we propose this to be the periplasmic side since the opposite side of the protein has considerably more positive charges (Fig. 5). Thus, the N- and C-termini extend into the periplasm. The protein includes a pocket or cavity extending halfway across the membrane that is open to the periplasm (Fig. 6a). The pocket also opens to the periplasmic leaflet of the membrane where H4 and H8, with kinks at strictly conserved Pro101 and Pro202, respectively, cross one another and splay apart. As discussed below, the active site of the enzyme is proposed to reside at the bottom of this pocket. The N-terminal end of h7 creates the base of the pocket next to which sits a lobe of electron density (Fig. 6d). Short amphiphilic helices exist in the loops connecting H3 and H4 (H3′) and H4 and H5 (H4′). Based on a Dali search[23], the fold is unique. The vast majority of residue pairs predicted by the co-evolutionary method are proximal to each other in 3D space within the structure (Supplementary Fig. 4)[24].

The protein fold in BacA is of the interdigitating inverted-topology repeat (IITR) type, as predicted[24] (Fig. 3, Supplementary Fig. 5). It consists of an N-repeat (residues 1–150) and a C-repeat (residues 151–273). Individual repeats begin with a re-entrant helix pair followed by three transmembrane helices. Each inserts into the membrane with opposite orientation and is related to the other by an approximate twofold rotation ($C_2$) axis in the plane bisecting the membrane. The symmetry axis passes through the center of both repeats. As a result, individual repeats possess helices that interdigitate making for a complicated folding pattern reminiscent of domain-swapping. The last three sequential helices in the C-repeats and N-repeats are arranged around the protein core in clockwise and counter-clockwise directions, respectively (Fig. 3d). Interdigitation gives rise to sequential helices that are widely separated in the membrane plane. Because the two repeats have limited sequence homology (16% identity, 36% similarity) (Supplementary Fig. 1) and a less than complete structural homology (Fig. 7), pseudosymmetry best describes the similarity in the backbone fold or topological arrangements they share.

Whilst the folding pattern in BacA is complicated, it can be viewed simply as consisting of two structural domains (Figs 3a, 4). Domain 1 includes h1, h2, H3, H9 and H10. Domain 2 includes H4, H5, h6, h7 and H8. In domain 1, h1 and h2 together take the form of an arch (h1–h2) that sits within an outer arch H9–H10 (Fig. 4b). This 'double arch' motif (double arch 1) is repeated in domain 2 with h6–h7 and H4–H5 as inner and outer arches (double arch 2), respectively (Figs 3a and 4d). Each domain therefore consists of a double arch and what we refer to as a buttress helix, H3 and H8 in the case of domains 1 and 2,

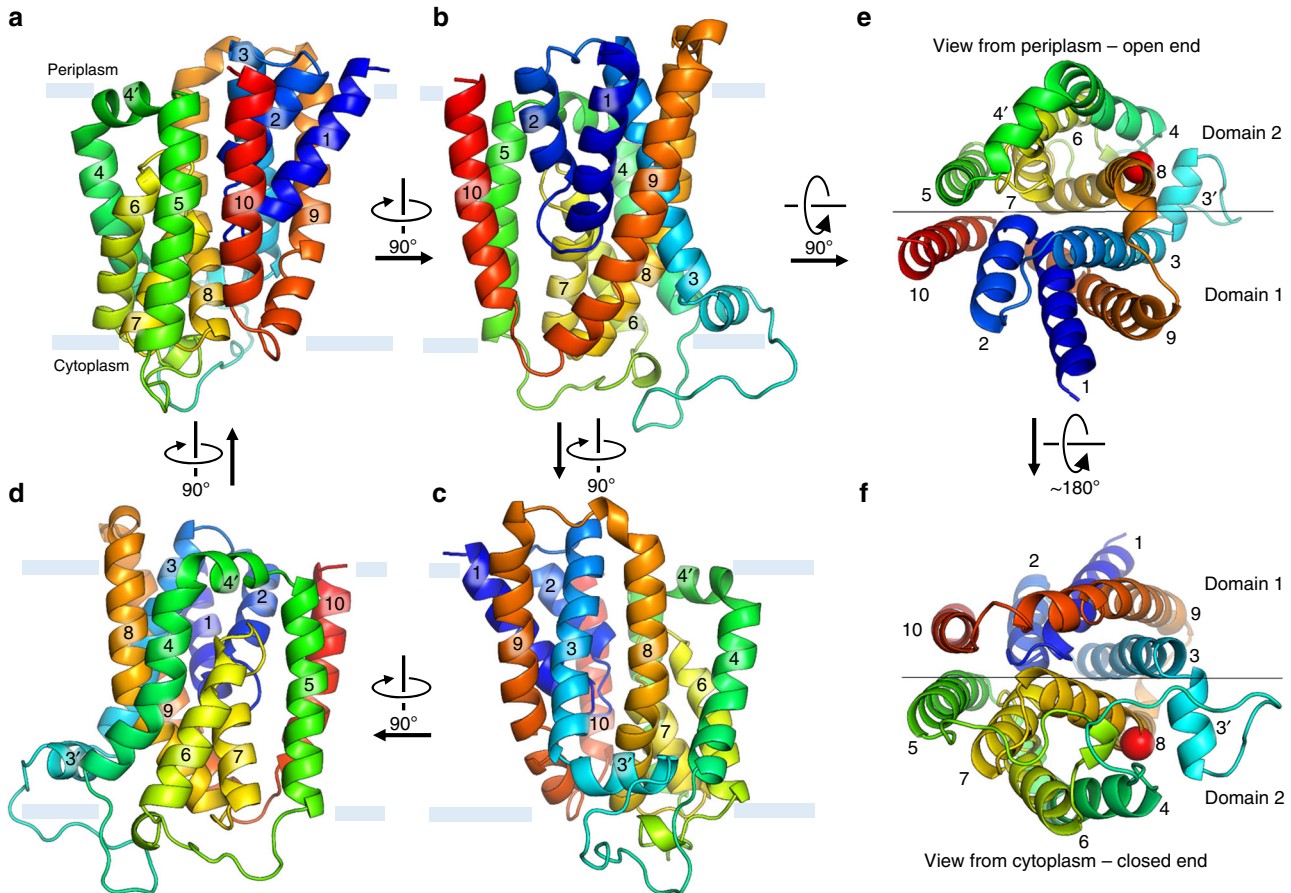

**Fig. 4** Overall architecture of BacA from *E. coli*. **a–d** Views of BacA from within the membrane rotated by ~90 degrees about an axis normal to the membrane plane and passing through the core of the protein. **e, f** Views of BacA from the periplasm/open and cytoplasm/closed end, respectively. The thin black line in **e**, **f** marks the location of the approximate C₂ axis of pseudosymmetry and separates BacA into two domains. Pro202 at the kink in H8 is shown as a red sphere

respectively (Fig. 4c). Contacts between the two domains involve H3 next to H8 and H5 next to H10 arranged as antiparallel helices across the interface (Fig. 4a, c). Short helices h2 and h7 meet at the putative active site with N-terminal ends apposed and helical dipoles opposed (Fig. 6d–f). Importantly, the two structural domains consist of swapped segments from the N- and C-repeats (Figs 3d, 4e, f). Thus, domain 1 includes segments h1–h2–H3 of the N-repeat and H9 and H10 of the C-repeat. Domain 2 includes segments H4 and H5 of the N-repeat and h6–h7-H8 of the C-repeat (Figs 3c, d and 4e, f). The domains are related to one another by the same approximate twofold axis of rotational symmetry that lies in the mid-plane of the membrane as pertains to the N-repeats and C-repeats (Fig. 3a).

**Putative active site**. The cavity or pocket that presumably contains the active site of BacA resides at the interface between the two domains. The domains are related to one another by a 180 degree rotation about an axis in the mid-plane of the membrane. However, because the domains are not identical structurally, when viewed from either side of the membrane the two ends of the protein are different. Specifically, one has an open pocket, the other does not and appears closed (Fig. 4). The distinction comes about primarily due to a difference in the position relative to the protein core of the cytoplasmic and periplasmic halves of symmetry related H3 and H8, respectively. In turn, this arises because H3 is a straight helix. By contrast, H8 is kinked at its middle at

Pro202 with an obtuse angle of ~145 degrees (Fig. 4e, f). As a result, while its periplasmic half is oriented approximately parallel to the membrane normal, effectively creating an open pocket, its cytoplasmic half bends toward the protein core, blocking and closing any potential pocket. Indeed, reference points at the ends of H3 and H10 on either side of the pocket are 21 Å apart at the open end of the protein. At the closed end, the corresponding distance between H8, equivalent to H3 at the open end, and H10 is 9 Å (Fig. 4e, f).

For the most part, conserved BacA1 and BacA2 motif residues map onto the pair of re-entrant helices (Supplementary Fig. 1, Supplementary Fig. 6). The bulk of these decorate the ends of the four helices and the loop between helix pairs where they come together toward the mid-plane of the membrane. We speculate that the active site of the enzyme resides in this region of the protein which, as noted creates a pocket open to the periplasm and to the periplasmic leaflet of the membrane (Fig. 6a). Extensive mutagenesis studies support this proposed placement (Supplementary Table 1)[18,19]. The N-terminal end of h7 resides at the base of the pocket. It seems likely therefore that the C55PP substrate enters the active site from the membrane periplasmic leaflet through the H4/H8 cleft, pyrophosphate head group at the membrane interface first, with the polyisoprenyl tail trailing behind in the hydrocarbon recesses of the bilayer. The L1–2 and L6–7 loops have their peptide amide hydrogens pointing into the binding pocket. The partial positive charge associated with peptide amides plus the guanidinium moieties of nearby Arg174

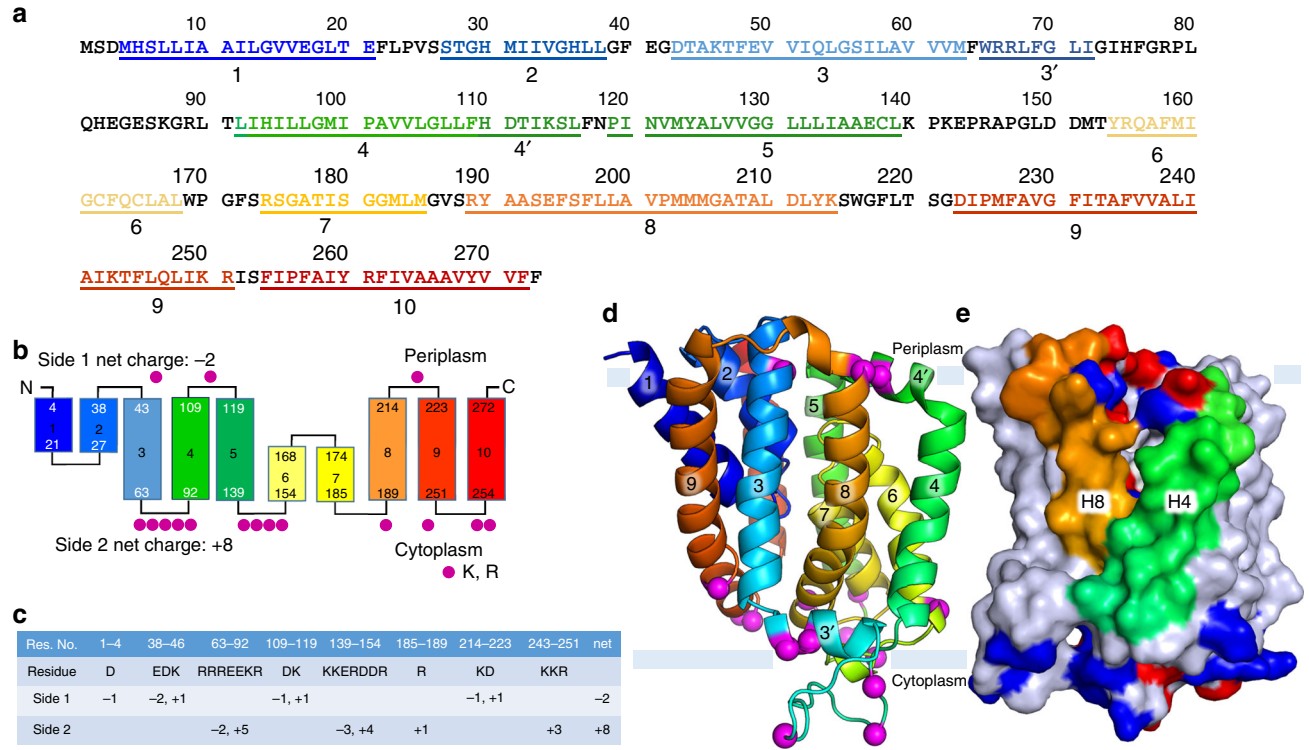

**Fig. 5** Positive-inside rule tally for BacA. **a** Sequence of BacA from *E. coli* rainbow color coded from N-terminus to C-terminus. Helices observed in the crystal structure are numbered as described in the text. **b** Distribution of cationic residues arginine and lysine in the periplasmic and cytoplasmic exposed stretches of polypeptide. Arginines and lysines on the cytoplasmic side considerably outnumber those on the periplasmic side of the membrane suggesting the protein is oriented with its N-terminus and C-terminus in the periplasm. **c** The table enumerates the relevant amino acids on either side of the membrane. **d** Overall architecture of BacA indicating the cationic residues, as described in **b**, as magenta spheres. **e** Surface representation of BacA where cationic (Arg, His, Lys) and anionic (Asp, Glu) residues are colored blue and red, respectively. H4 and H8 are colored green and orange, respectively. All other helices are colored gray

and Arg261 collectively should serve to favor interaction with the anionic pyrophosphate head group of the substrate thereby positioning it in the active site for hydrolysis (Figs 6e and 8). MD simulations (MDS) of C55PP complexed with BacA support this proposed interaction (Supplementary Fig. 7).

In an attempt to capture a structure of BacA with either substrate or product bound, crystallization trials were set up with the enzyme pre-incubated with either C55PP or C55P. To prevent pyrophosphorolysis in the case of C55PP, the inactive mutant S27A was used. Wild-type BacA was employed with C55P. In both cases, pre-incubation was performed in the presence of calcium, the divalent cation with which BacA is most active[18,19]. In addition, trials were carried out using mesophase doped with C55P in an effort to fully saturate the enzyme with product and to favor crystallization of the BacA-C55P complex. A similar strategy for obtaining ligand complex structures has been implemented successfully with human mPGES1 and ArnT[25,26]. Despite these extensive efforts no convincing extra electron density was observed in the proposed binding pocket that could be ascribed to an added ligand. In all cases, a small lobe of density in the pocket was present, as already noted (Fig. 6d).

The enzyme has a requirement for divalent cations and is most active with calcium[18,19]. Thus, calcium may enter the active side already complexed to the pyrophosphate of C55PP. Alternatively, calcium resides in the active site chelated in place by conserved Glu17 and Glu21 for use in stabilizing the head group of C55PP for pyrophosphorolysis. Indeed, docking of calcium into the active site is suggestive of a binding site coordinated by the two acidic residues. Ser27 is a strictly conserved residue which upon mutagenesis to Ala reduces the pyrophosphatase activity of BacA

with C55PP as substrate by 7550-fold compared to wild type (Supplementary Table 1)[18]. Ser27 has been speculated to provide the nucleophilic hydroxyl with which to attack the β-phosphate of C55PP and the residue that becomes phosphorylated ahead of cleavage by a catalytic water. Ser27 resides at the N-terminus of h2 toward the base of the active site pocket within 7 Å of Glu17 and Glu21 where the β-phosphate is expected to sit in the Michaelis complex (Fig. 8, Supplementary Movie 1, Supplementary Movie 2). Adjacent conserved Ser26 and Thr28 and the backbone amides on either side of Ser27 may contribute to enhancing the nucleophilicity of Ser27. Indeed, the mutant Ser26Ala retains just 15% of wild-type activity[18]. Upon pyrophosphorolysis, the inorganic phosphate product would likely leave the active site via the opening to the periplasm. A number of basic residues (His36, Lys46, Lys114) are located around the periplasmic rim of the pocket to facilitate the process. The C55P product would presumably exit in the same way that C55PP entered the site via the H4/H8 cleft thereby resetting the enzyme for a new round of catalysis.

**Alternate states**. IITRs have been described in channels and transporters; LeuT is a prime example[27,28]. The asymmetry that naturally exists in such proteins enables alternating access and vectorial movement of substrates across the membrane. Thus, conformational swapping between the two repeats induces a switching between inward-facing and outward-facing states. The attendant asymmetry exchange in which each repeat alternates between two conformations allows the protein to change from being asymmetric and open on one side of the membrane to

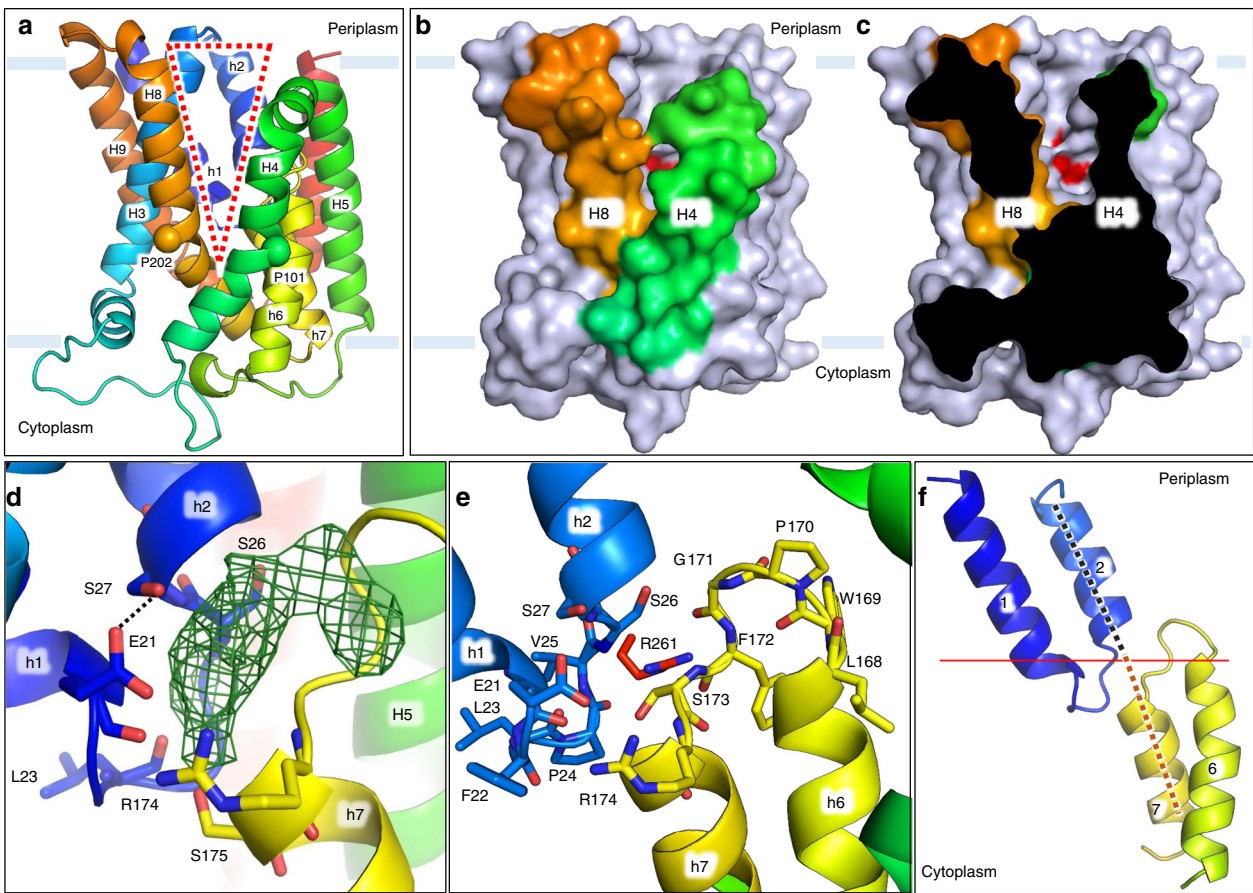

**Fig. 6** Active site pocket of BacA. **a** View into the active site from the membrane through the cleft between H4 and H8. The opening from the membrane is marked by a dashed triangle. Access from the periplasm is in the opening marked by the base of the triangle where the helices splay apart. **b** Surface representation of the view into the active site shown in **a**. Ser27 in the active site is colored red. **c** Slice through the view in **b** to reveal the active site pocket with access to the periplasm. **d** Electron density in the active site. The Fo–Fc omit density map (green mesh) contoured at 2.5$\sigma$ shows a lobe of density in the proposed substrate binding pocket. The density sits approximately equidistant from the apposed N-termini of h2 and h7 on the axis of symmetry that passes through the protein in the membrane plane. Attempts made to fit the density with phosphate, pyrophosphate, glycerol, calcium, ammonium, citrate, DDM and water molecules proved unsatisfactory. The distance between E21 and S27 is 3.1 Å (dashed line). **e** Details of the proposed binding pocket for the pyrophosphate head group of C55PP. The binding pocket includes the guanidinium moieties of Arg174 and Arg261 and the ring of partial positive charges from the amide backbone of loops L1–2 (E21-S27) and L6–7 (L168-R174). Residues in loops are shown as sticks. Polypeptide in the foreground (H3 and H8) and background (H5 and H10), excluding residue R261 in H10, has been removed for clarity. **f** Alignment of helical axes in helices h2 and h7 on either side of the putative active site. The N-termini of h2 and h7 are ~9 Å apart with helical dipoles opposed. Parts of the protein to the forefront and background have been removed for clarity. The horizontal line marks the location of the approximate $C_2$ axis of pseudosymmetry. The dashed lines align with the helix axis of h2 and h7

assuming a second asymmetric state and open on the other. Interestingly, a structure of one of the two limiting states can be used to homology model the alternate state after threading one repeat sequence into the other[27,28]. This approach has been implemented successfully with BacA. As expected, the alternate conformation is open to the cytoplasm and to the cytoplasmic leaflet and closed at its periplasmic end (Supplementary Fig. 8). As with the outward open conformation captured in the crystal structure, this modeled inward open state is stable in silico in a simulated hydrated membrane.

It is instructive to examine changes in and around the putative binding pocket in the morphing sequence between the crystal structure or open form and the homology modeled closed state (Supplementary Movie 3, Supplementary Fig. 8). In the former, the binding pocket is open to the periplasm and to the periplasmic leaflet. The closed state forms as the N-terminal stretch of H3 moves over the binding pocket closing it off to the periplasm. Simultaneously, H4 straightens and elongates becoming continuous with H4′. In so doing, this remodeled helix moves

toward H8 to seal the cleft between the binding pocket interior and the outer leaflet of the membrane. The impression given by the morphing sequence is that the closing process corresponds to the final stages of the pyrophosphatase reaction where the products C55P and inorganic phosphate are expelled from the binding pocket. A reopening of the pocket, as revealed in the morphing sequence, would reset the enzyme for another round of catalysis. At no point in the cycle does a continuous pore through the enzyme form. Thus, the integrity of the membrane is maintained throughout the process.

**Structural changes during catalysis**. That the catalytic cycle involves large structural changes in BacA was tested experimentally by disulphide cross-linking. For a disulphide bond to form between two cysteines in a protein the corresponding $C_\alpha$ atoms must come within 4 to 9 Å of one another. The morphing sequence showed that loop L2–3 and amphiphilic helix H4′ on the periplasmic end of the protein move noticeably toward one

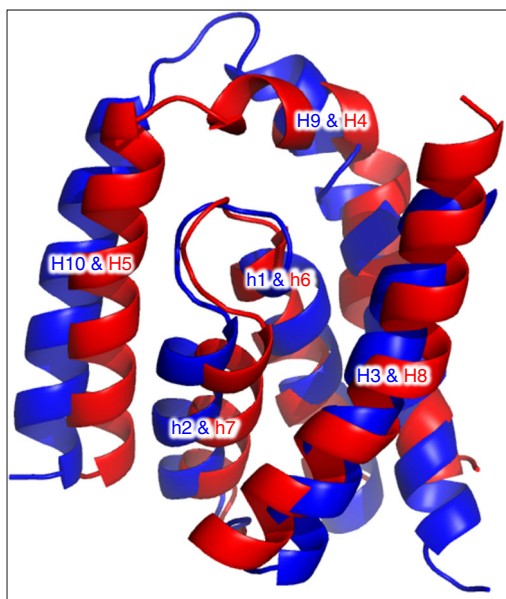

**Fig. 7** Superposition of domains 1 and 2 in BacA. The $C_\alpha$ r.m.s.d between superposed domain 1 (blue) and domain 2 (red) over 108 residues is 2.34 Å. The view is in the membrane plane

another upon binding pocket closure. Indeed, Gly42 on L2–3 and Phe117 on H4′ with $C_\alpha$s 13.6 Å apart in the open state are separated by 7.4 Å in the closed state (Supplementary Fig. 8). To test the validity of the morphing sequence and its relevance to catalysis, the Gly42Cys/Phe117Cys double mutant was prepared and the pyrophosphatase activity of the new construct was measured with and without the reducing agent, β-mercaptoethanol (βME). The mutant protein, purified under oxidizing conditions, had an activity that was 2% of wild type (Supplementary Table 1). This is consistent with disulphide bond formation between the two cysteines and an immobilization of the periplasmic end of the protein akin to the modeled closed state that dramatically compromised enzymatic activity. In the presence of βME where the two cysteines should be fully reduced and the protein unconstrained, enzyme activity was 71% of that recorded for the wild-type protein. These data suggest therefore that at some point in the catalytic cycle the two mutant cysteines come together to enable cross-linking and that flexibility and movement, at least at the periplasmic end of the protein, are directly relevant to BacA catalysis.

## Discussion

BacA is an enzyme responsible for the phosphoanhydride hydrolysis of C55PP. The assumption is that the reaction is catalyzed on the periplasmic side of the membrane providing C55P in support of cell wall component synthesis intracellularly. An obvious question arises as to the purpose served by the IITR fold upon which BacA is built. As noted, IITRs have been identified and their function rationalized in channels and transporters. To our knowledge, they have not been reported for enzymes. IITRs are thought to originate as a result of gene duplication and subsequent fusion. Why this might have occurred in the case of BacA remains a mystery. However, given the close links between IITRs and transporters, it is worth considering if BacA has any transport, channel, or flippase activity. In this regard, it is possible that the product of the pyrophosphatase reaction, C55P, gets transported across the membrane by BacA for re-entry into the PCS cycle on the cytoplasmic side (Fig. 1). One argument against

this possibility however relates to the homology modeled alternate inward open state where the cleft appears between H3 and H9 (Supplementary Fig. 8). In the outward open state, observed in the crystal structure, the cleft resides between H4 and H8. It is between these two helices that the long polyisoprene tail of the substrate and product is proposed to extend during binding and catalysis (Supplementary Fig. 8). For BacA to act as a flippase, by a simple rocker-switch mechanism, the cleft between H4 and H8 would shift along the length of the two helices. This is not consistent with the alternation model where the cleft that forms on the other side of the membrane resides between H3 and H9 in the inward open conformation (Supplementary Fig. 8).

Whilst the focus to date has been on the cleft between H4 and H8, one other possibility merits consideration. That is the space between transmembrane helices H5 and H10 in domains 2 and 1, respectively (Fig. 3a). H5 and H10 are extremely hydrophobic and are in close contact along their full length. A small separation of the two helices at their periplasmic halves would suffice to provide access to the active site for the C55PP substrate. An attractive feature of this scenario is that extending the separation between the two helices along their full length would provide a route for the C55P product to exit the binding pocket on the cytoplasmic side of the membrane. In this way, BacA would function seamlessly as both an enzyme and a flippase. One argument against this model is the strong interaction between the guanidinium of Arg261 on H10 and backbone carbonyls of Ala125 and Ile126 in H5, Pro24 in loop L1-L2 and Phe172 in loop L6-L7 (Supplementary Fig. 9). For the model to apply, these interactions would have to be broken. The anionic head group of C55PP interacting directly with Arg261 could well facilitate the process. With the polyisoprene tail of the substrate or product seated between the separated helices at all times during the process, an opening across the membrane is avoided.

The twofold axis of (pseudo)symmetry passes through the interdigitated N-repeats and C-repeats and through the base of the binding pocket where the re-entrant helices meet. The two short helices h2 and h7 from opposite sides of the membrane are aligned with their helical long axes close to coincident and with their helical dipoles opposed; the N-termini are ~9 Å apart (Fig. 6f). Interestingly, the lobe of density in the active site sits next to the space between and equidistant from the two termini (Fig. 6d). As noted in cases of other proteins displaying inverted twofold pseudosymmetry, the symmetry element typically demarks a functional part of the protein where substrates bind and/or pathways reside[27,28]. This again, suggests that in BacA the observed pseudosymmetry is there for a reason; it serves some purpose, as speculated on above.

Inverted twofold pseudosymmetry has been proposed to originate from gene duplication. In this model, the ancestral gene codes for a monomeric protein that is functional, stable and can orient both ways across the membrane (Fig. 9). The two evolve separately and eventually fuse as a result of gene fusion. The longer the evolutionary time the more likely it is that the two repeats can diverge in sequence and conformation. This would appear to have been the case with BacA, most noticeably at the sequence level. To what selective pressures are the two repeats responding in the membrane? One response is likely to reflect the profoundly different compositions and environments in and next to the inner and outer leaflets of the cell membrane.

Another obvious question is, what functionality could a repeat in isolation possess and express? And what is the driving force for existing in a fused state with inverted twofold pseudosymmetry—other than to enable pyrophosphorolysis? It is always possible that subsequent to the fusion event, the two repeats evolved with one repeat and, by extension one active site, becoming degenerate. This type of evolved functional asymmetry is not without

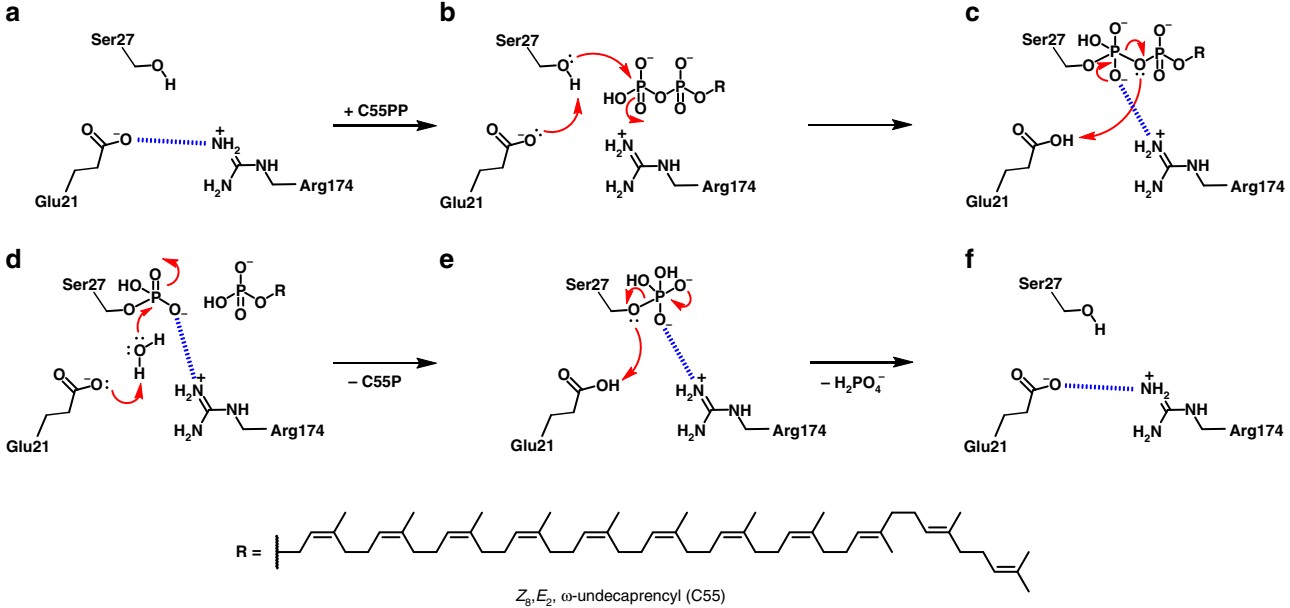

**Fig. 8** Proposed undecaprenyl-pyrophosphate phosphatase reaction mechanism in BacA. **a** Empty active site. Glu21 and Arg174 form a salt bridge as indicated by a blue dashed line. **b** Michaelis complex with C55PP. Glu21 serves as a proton acceptor that facilitates the nucleophilic attack of the hydroxyl group of Ser27 at the β-phosphate of C55PP. **c** Collapse of the pentavalent intermediate, phosphorylation of Ser27 and cleavage of the pyrophosphate bond. Upon disruption of the salt bridge between Glu21 and Arg174, the latter can stabilize the pentavalent intermediate in close proximity via ionic interactions. **d** Product complex. Hydrolysis of the phosphate group at Ser27 is mediated by a water molecule. Stabilization of the pentavalent intermediate by Arg174 potentially facilitates the exit of the product C55P from the active site. **e** Collapse of the second pentavalent intermediate. **f** Empty active site ready to undergo another reaction cycle. Calcium likely plays a role in catalysis by coordinating with the polar head groups of C55P and C55PP and the pyrophosphate pentavalent intermediate. Electron lone pairs are shown as double dots. Red curved arrows indicate electron flow. Dashed blue lines denote ionic interactions. R represents the polyisoprenyl chain of C55 with the double bond configuration of bacterial $Z_8,E_2,\omega$-undecaprenol (structure in inset at bottom). See also Supplementary Movie 1 and Supplementary Movie 2

precedent being evident in eukaryotic fluoride channels[29]. It is also possible that BacA evolved from an ancestral protein that had channel or transporter activity to the point where it no longer facilitates the movement of matter across the membrane.

Inverted twofold pseudosymmetry is used in channels and transporters to enable alternating between inward-facing and outward-facing states, by relative rocker-switch-like motion of two domains in the simplest case. An intriguing possibility exists that something like this might happen with BacA. Thus one could imagine an alternation between two states. In one, the enzyme functions selectively with substrate coming from the periplasmic leaflet. In the other, it is active with substrate from the cyto-plasmic leaflet only. Thus, toggling between the two could be as simple as a slight rotation of the two domains (interdigitated) with respect to one another about some pivot point or axis to open access to and egress from the active site/s from alternate bilayer leaflets. Control over which side the enzyme is active on might be dictated by the composition and aqueous environment

of each leaflet. An extension to this idea concerns the nature and activity of the two active sites. It might be that the enzyme shows differential substrate specificity depending on the side from which the substrate enters the active site or more specifically which active site is used. Thus, C55PP may be the preferred substrate for the periplasmic facing active site. Whilst another substrate is preferred at the cytoplasmic facing site.

BacA's intricate interdigitated inverted topology raises interesting questions about how such a complex structure folds in the membrane during and/or post-synthesis. One possible mechanism is outlined here (Supplementary Fig. 10). Ribosomal polypeptide synthesis proceeds directionally from N-terminus to C-terminus. Thus, the first elements in BacA to emerge are N-terminal helices h1 and h2. In the mature protein, these exist as a re-entrant helix pair connected by a loop. It is energetically unfavorable for a loop to reside alone in the apolar interior of a membrane. It is possible then that these either partition to the membrane interface or they combine to form one long metastable

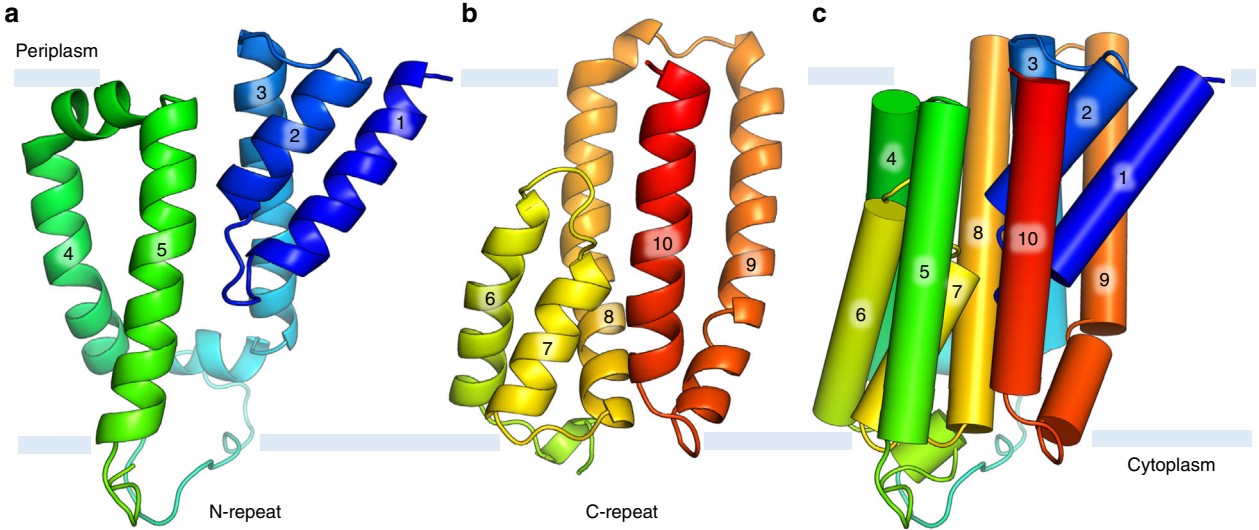

**Fig. 9** N-repeat and C-repeat in BacA viewed in isolation from within the membrane. **a** N-repeat. **b** C-repeat. Repeats include sequentially a re-entrant helix pair (the inner arch), a buttress helix and an outer arch. Repeats are shown with the same orientation they assume in the intact BacA. **c** Topology of BacA in simplified cylinder representation to illustrate how the N- and C-repeats in **a** and **b** interdigitate in the intact protein. It is possible that early in evolutionary time the individual N-repeat and C-repeat, in isolation, had functionality and stability. One would wonder if such isolated repeats could possibly be stable in a membrane with TM helices so far apart (to accommodate interdigitation in the intact, fused BacA protein). On inspection, transmembrane helices H3–H5 and H8–H10 are extremely hydrophobic along their full length suggesting that they could indeed be stable, as disposed in BacA, as isolated repeats or domains. But if isolated, they could presumably fold together reversibly in the membrane to form a helical bundle. Another question arises immediately, What happens to the re-entrant helices under these conditions? They too are extremely hydrophobic. However, the re-entrant loops L1–2 and L6–7, whilst not having many polar residues (FLPVSS in L1–2, WPGFS in L6–7), have peptide linkages in need of hydrogen bonding partners and thus are unlikely to be stable in isolation buried in the membrane's apolar interior. One solution would be for each of the re-entrant helices to form a long transmembrane

helix (H1H2) that spans the membrane. Given the highly apolar nature of h1 and h2, a single transmembrane helix is a more likely outcome. Polypeptide synthesis continues with H3, H4 and H5 being fed sequentially across the membrane. In so doing, H4 and H5 create the outer arch of domain 2 which serves to accommodate nascent inner arch h6–h7. Synthesis of H8 completes the assembly of domain 2 which consists of sequential helices H4, H5, h6, h7 and H8. With the emergence of H9 and H10, the outer arch of domain 1 is formed. Metastable H1H2 can now refold beneath H9–H10 creating inner arch h1–h2 and a fully formed domain 1. The last step in the proposed process involves the two domains folding toward one another about an axis between and parallel to H3 and H8. This operation keeps H3 next to H8 and brings H5 and H10 together with the helix axis of h2 and h7 coincident and with their N-termini juxtaposed at the core of the putative active site. Stabilization of the paired domains involves the guanidinium of Arg261 in the middle of H10 engaging with four backbone carbonyls in H5, L1–2 and L6–7 (Supplementary Fig. 9). This locks the domains together creating an enclosed active site and functional BacA.

## Methods

**Materials.** Tris-HCl (Cat. 5429.3) was purchased from Roth (Karlsruhe, Germany), sodium chloride (Cat. 7647-14-5) was purchased from Fisher Chemical (Loughborough, UK), and glycerol (Cat. 6782-25) was obtained from MACRON (PA, USA). $n$-Dodecyl-β-D-maltopyranoside (DDM) (Cat. D310) and $n$-decyl-β-D-maltopyranoside (DM) (Cat. D322) were purchased from Anatrace (OH, USA). Isopropyl-β-D-thiogalactopyranoside (IPTG) (Cat. OP-0010) was purchased from Eurogentec (Liège, Belgium). Monoolein (9.9 MAG) (M239) was from Nu-Chek Prep (Elysian, MN). PEG 400 (Cat. 81172), β-mercaptoethanol (βME, Cat. M6250), farnesyl-pyrophosphate (C15PP) (Cat. F6892) and sodium citrate tribasic dihydrate (Cat. 71402), were obtained from Sigma (St. Louis, MO). Ammonium citrate dibasic (Cat. HR2-245) and mercury (II) chloride (Cat. HR2-446) was purchased from Hampton Research (Aliso Viejo, CA). Undecaprenyl-monophosphate di-ammonium salt (C55-MPDA) (Cat. 62-1055-2) was purchased from LGC Standards (Molsheim, France) and undecaprenyl diphosphate triammonium salt (C55PP) (Cat. 12-11-4) was sourced from the Polish Academy of Sciences. [14C] Isopentenyl-pyrophosphate (C5PP) (Cat. NEC773010UC) was purchased from

Perkin Elmer (MA, USA). Microsyringes (Cat. 81030) were sourced from Hamilton (Bonaduz, GR, Switzerland). Cyclic olefin copolymer (COC) (Cat. TOPAS 8007) and harvesting cryo-loops (Cat. M2-L18SP-20, M2-L18SP-30 and M2-L18SP-50) were obtained from MiTeGen (Ithaca, NY). The goniometer base (Cat. MD7–400, CryoCap (SPINE standard) was purchased from Molecular Dimensions (Florida, USA). Standard glass (127.8 × 85.5 mm², 1 mm thick; Cat. 1527127092) and No. 1.5 glass (124 × 84 mm², 0.15 mm thick; Cat. 01029990933) were obtained from Marienfeld (Lauda-Königshofen, Germany). Perforated double-stick spacer tape (112 × 77 mm²; 140 µm thick; perforations, 6 mm diameter) (Cat. 9500PC and 9009), and double-stick gasket (2 mm wide and 140 µm thick with outer dimensions 118 × 83 mm² and inner dimensions 114 × 79 mm²; TRI-9500PC) were purchased from Saunders (MN, USA). Glass cutting tools (TCT Scriber & Glass Cutter, Cat.180714) were obtained from Silverline (Yeovil, UK). Rain-X rain repellent (Cat. 80199200) was from Shell Car Care (Altrincham, Cheshire, U.K.).

**Plasmids and site-directed mutagenesis.** The previously described expression vector pTrcBac30 containing the wild-type gene *bacA* from *E. coli*[6] or the S27A mutant gene[18] were used for the expression of BacA and BacA S27A proteins. Each recombinant construct contains a His$_6$-tag at the N-terminus (Met-His$_6$-Gly-Ser extension). BacA mutants were obtained via PCR using the Q5 Site-Directed Mutagenesis Kit (cat. M0492S) from New England Biolabs (USA) with the appropriate primers whose sequences are reported in Supplementary Table 2. Mutations were introduced directly in the pTrcBac30 expression plasmid and the entire *bacA* gene insert was confirmed by sequencing.

**BacA expression and purification.** The pTrcBac30 recombinant plasmids were transformed into chemically competent *E. coli* C43(DE3) (Avidis-France) cells. Cells were grown in TB broth (Sigma, St Louis, MO), supplemented with ampicillin (100 µg mL$^{-1}$) (Melford) at 37 °C to an OD$_{600}$ of 1 and then induced with 1 mM IPTG at 22 °C and 180 x g for 16 h. Cells were harvested by centrifugation at 4000 x g for 20 min at 4 °C. Cells were either used immediately or stored at −20 °C for a maximum of 2 months. About 15 g of BacA cell mass was re-suspended in 40 mL of lysis buffer (25 mM Tris-HCl pH 7.2, 150 mM NaCl, 20 %(v/v) glycerol, 2 mM βME (omitted with double cysteine mutant), 2 mM MgSO$_4$, 1.5 U mL$^{-1}$ benzonase). Cells were lysed by three passages through a high-pressure homogenizer (Emulsiflex-C5; Avestin). A membrane fraction was separated from the supernatant by ultra-centrifugation at 100,000 x g for 1 h at 4 °C. The membrane pellet was re-suspended in 40 mL of solubilization Buffer A (25 mM Tris-HCl pH 7.0, 150 mM NaCl, 20 %(v/v) glycerol, 2 mM βME, 1 %(w/v) DDM) and incubated for 2 h at 4 °C under gentle agitation before being centrifuged at 4 °C for 1 h at 100,000 x g. The supernatant containing mostly contaminants was discarded and a second extraction/solubilization was performed similarly by re-suspending the pellet with

Buffer A containing 1.5 %(w/v) DDM. The recovered supernatant containing most of the expressed BacA protein was filtered through a 0.22 μm membrane (Millex-GP, Millipore) and loaded on a 1 mL Histrap column (GE Healthcare) equilibrated with Buffer B (25 mM Tris-HCl pH 7.0, 150 mM NaCl, 5 %(v/v) glycerol, 2 mM βME, 0.05 %(w/v) DDM). The column was washed with 60 mL of Buffer B containing increasing imidazole concentrations (10 mM to 40 mM). Bound protein was eluted with 15 mL of Buffer B containing 200 mM imidazole. Fractions containing BacA were combined and dialyzed for 2 h against 40-fold volumes of Buffer B without DDM. The dialyzed protein was concentrated to 1.5 mL using a 15 mL Sartorius centrifuge filter with a molecular weight cutoff of 30 kDa. The concentrated sample was loaded onto a 5/150 GL column packed with Superdex 200 resin pre-equilibrated with Buffer B at 4 °C using a GE Healthcare AKTA Purifier system. Size-exclusion chromatography was performed at a flow rate of 0.5 mL min$^{-1}$. BacA was eluted as a single, symmetrical $A_{280}$ peak. For the crystallization experiments, the DDM detergent was replaced by DM using an additional Ni-NTA purification step. Purified protein was loaded onto a 1 mL Histrap column (GE Healthcare) equilibrated with Buffer B containing 0.05 %(w/v) DDM. The column was washed with 50 mL of Buffer C (25 mM Tris-HCl pH 7.2, 150 mM NaCl, 20 % (v/v) glycerol, 2 mM βME, 1 %(w/v) DM). BacA was eluted with 200 mM imidazole in Buffer C. The eluted protein was desalted using a PD-10 desalting column (GE Healthcare) using Buffer C. BacA was concentrated to 12 mg mL$^{-1}$ using 30 kDa cutoff concentrators from Sartorius. Purification was monitored by SDS-PAGE. Protein concentration was determined by measuring the absorbance at 280 nm using an extinction coefficient of 25,565 M$^{-1}$ cm$^{-1}$ (http://web.expasy.org/protparam/). The homogeneity of the sample was verified by dynamic light scattering using a Wyatt Dynapro Nanostar system.

**C55PP phosphatase assay.** The C55PP phosphatase assays were performed in a 10 μL reaction mixture containing 20 mM Tris-HCl pH 7.4, 2 mM CaCl$_2$, 10 mM βME (omitted with double cysteine mutant under oxidizing conditions), 150 mM NaCl, 0.2 %(w/v) DDM and 50 μM radiolabelled [$^{14}$C]C55PP (900 Bq), which was synthesized and purified as previously described[6]. The BacA concentration was adjusted to achieve <30 % substrate hydrolysis. The reaction was incubated for 10 min at 37 °C and subsequently stopped by freezing in liquid nitrogen. Substrates and products were separated and quantified by thin layer chromatographic analysis on precoated silica gel 60 (Merck) plates using diisobutyl ketone/acetic acid/water (8:5:1 by vol.) as the mobile phase. Radioactive spots were located and quantified with a radioactivity scanner (model Multi-Tracermaster LB285, Berthold-France)[6].

**Crystallization.** To obtain crystals suitable for SAD phasing purified BacA at 12 mg mL$^{-1}$ in Buffer B was incubated for 10 min on ice with 2 mM mercury (II) chloride. This solution was then used to prepare the cubic phase with monoolein as host lipid using a coupled-syringe mixing device[20]. In meso in situ serial crystallization (IMISX) trials were set up by transferring 50 nL of the protein laden mesophase onto a siliconized 96-well cyclic olefin copolymer (COC) crystallization plate which was subsequently overlain with 800 nL precipitant solution using an in meso robot. Plates were covered with a COC cover plate, sealed within a glass sandwich plate and incubated and imaged at 20 °C with a Formulatrix Rock Imager (RockImagerRI1500, Formulatrix, Inc, Waltham, MA). Crystallization progress was monitored manually using bright field and polarized light microscopy (Eclipse E400 Pol, Nikon, Melville, NY). Best quality crystals were obtained using precipitant solutions containing 40 %(v/v) PEG-400, 0.3–0.5 M ammonium citrate dibasic and 0.1 M sodium citrate pH 5.0. Plate shaped crystals appeared after 3 days and continued to grow reaching dimensions of $2 \times 15 \times 20$ μm$^3$ in 3 weeks.

**Data collection.** X-ray diffraction experiments were carried out on protein crystallography beamline X06SA-PXI at the Swiss Light Source, Villigen, Switzerland. Data were collected at 100 K using the IMISXcryo method[21]. Briefly, this involves growing crystals in double sandwich plates and collecting diffraction data directly on crystals where and as they grow in the inner plate without the need for mesophase handling or crystal harvesting. Measurements were made in steps of 0.1° at 0.1 s per step with the EIGER 16 M detector operated in a continuous/shutterless data collection mode at a sample-to-detector distance of 15 cm. Beam size was adjusted to $10 \times 20$ μm$^2$ to match the average crystal size for better signal/noise ratio. SLS data acquisition software, DA+, was used for fast two-dimensional grid scanning, semi-automated crystal picking and data collection[30]. BacA-Hg crystals were measured at wavelength and flux values of 1.9 Å and $4.4 \times 10^{10}$ photons/s, respectively. Data sets from two IMISX wells were used for experimental phasing and refinement. Of the 66 data sets collected, 36 came from well 1 recorded over a 30° wedge per crystal and 30 were from well 2 recorded over a 15° wedge per crystal.

**Data processing and merging.** Data sets with wedges of 30° (well 1) and 10° (well 2) per crystal were indexed and integrated using XDS[31], as previously described[21]. Of the 66 data sets recorded, 64 were integrated successfully and then scaled and merged with XSCALE to obtain a preliminary data set. Subsequently, 11 data sets with $IS_a$ values in the preliminary merging of <4 were rejected. The final data consisted of 53 data sets scaled and merged using XSCALE. Data collection and processing statistics are provided in Table 1.

**Structure determination and refinement.** Initial attempts to phase the diffraction data by molecular replacement using the predicted model of Ovchinnikov et al., (2015)[24] as template failed. Accordingly, the single-wavelength anomalous diffraction (SAD) method was employed for experimental phasing using the anomalous diffraction data set collected with crystals of BacA-Hg. Heavy-atom locations, structure phasing and density modification were performed using the HKL2MAP interface to SHELXC, SHELXD and SHELXE[32]. Two mercury sites were identified with 1000 SHELXD trials in the resolution range of 44.45 Å – 2.8 Å with CC$_{all}$/CC$_{weak}$ values of 49.3/27.6, and initial phasing employed 100 cycles of SHELXE density modification with autobuilding of the protein backbone trace. The starting model was built manually using Coot[33]. PHENIX[34] and BUSTER[35,36] were used for refinement. The final model was refined to 2.6 Å resolution with $R_{work}/R_{free}$ of 0.21/0.24. Refinement statistics are reported in Table 1. Figures were generated with PyMOL (http://www.pymol.org)[37].

The diffraction data and the refined model have been deposited in the Protein Data Bank under entry PDB ID 5OON.

**Molecular modeling and simulations.** Molecular models of the inverted-state of the BacA crystal structure were created by initially generating a structure-based sequence alignment between the N-repeat (residues 1 to 152) and the C-repeat (residues 153 to 273). Modeller[38] was used to model the invert-state based on this sequence alignment[28]. The topological orientation of BacA within the bacterial cytoplasmic membrane was predicted using sequence-based approaches TOPCONS[39] and MEMSAT-SVM[40] and the structure-based method Memembed[41].

GROMACS v5.1.2 was used to perform all molecular dynamics simulations (MDS)[42]. Coarse grained (CG) MDS with the Martini 2.2 force field[34,43] were used to perform initial 1 μs simulations to permit the assembly and equilibration of 1-palmitoyl-2-oleoyl phosphatidylglycerol (POPG):1-palmitoly-2-oleoyl phosphatidylethanolamine (POPE) bilayers around the BacA structures at a 1:3 mole ratio[44]. CG2AT was used to convert CG molecular systems to atomistic detail[45], with Alchembed applied to remove any unfavorable steric contacts between protein and lipid[46]. The atomistic systems equate to a total size of ~86,600 atoms and box dimensions in the region of $90 \times 112 \times 112$ Å$^3$. The systems were then equilibrated for 1 ns with the protein restrained, followed by 10 ns with only the backbone atoms restrained, before 3 repeats of 100 ns of unrestrained atomistic MDS, for each configuration of the molecular system (see below), using the Gromos53a6 force field[47]. Simulations were extended until 500 ns for the apo BacA X-ray structure. Molecular systems were neutralised with a 150 mM concentration of NaCl.

Gromos 53a6 force field parameters and coordinates for C55PP and C55P were downloaded from the Automated Topology Builder (ATB) Repository[48]. Phosphoserine parameters[49] were updated from Gromos 43a1 to the 53a6 force field. In both cases, the parameters were compared with other known phosphate topologies[50] and with bond lengths, angles and torsions. Partial charges were checked using Maestro (Schrödinger Release 2016–4: Maestro, Schrödinger, LLC, New York, NY, 2016). The pentavalent intermediate that forms between Ser27 and C55PP was modeled and parameterized based on an equivalent intermediate formed by diacylglycerol kinase, DgkA[50]. PROPKA[51] was used to estimate the pKa values of the titratable residues in the BacA structures.

All simulations were performed at 37 °C, with protein, lipids and solvent separately coupled to an external bath, using the velocity-rescale thermostat[52]. Pressure was maintained at 1 bar, with a semi-isotropic compressibility of $4 \times 10^{-5}$ using the Parrinello-Rahman barostat[53]. All bonds were constrained with the LINCS algorithm[54]. Electrostatics was measured using the Particle Mesh Ewald (PME) method[55], while a cutoff was used for Lennard-Jones parameters, with a Verlet cutoff scheme to permit GPU calculation of non-bonded contacts[56]. Simulations were performed with an integration time step of 2 fs. The MDS were analyzed using Gromacs tools, MDAnalysis[57] and locally written python and perl scripts.

Homologous sequences were identified in the UniProtKB database, using eight iterations of Jackhmmer[58], with the default search parameters. This identified 5564 non-identical BacA sequences. For the previous study that applied co-evolution constraints to fold the BacA structure with Rosetta, 4128 homologs were identified[24]. In a subsequent study, the authors identified 38,511 sequences from their metagenomics data set, of which 14,367 were non-identical[59].

For each sequence data set, the percentage conservation for each residue was mapped onto the residue B-factor column of the BacA X-ray structure. A Weblogo representation of the sequence alignment from the Jackhmmer search was created (Supplementary Fig. 1)[60].

**Data availability.** Data supporting the findings of this manuscript are available from the corresponding author upon reasonable request. Protein Data Bank accession numbers: The structure and structure factors for BacA from *E. coli* were deposited into the PDB under accession code 5OON.

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

## Acknowledgements

We thank past and present members of the Membrane Structural and Functional Biology group for their assorted contributions to the study. Dr Ahmed Bouhss is acknowledged for sharing his knowledge of the BacA protein. Drs. Charlier Paulette and Dominique Mengin-Lecreulx are recognized for their support of the project. The assistance and support of beamline scientists at the Swiss Light Source (X06SA and X10SA), Diamond Light Source (I24) and the Advanced Photon Source (23-ID) are acknowledged. The work was funded by Science Foundation Ireland grant 12/IA/1255, the Belgian program of Interuniversity Attraction Poles initiated by the Federal Office for Scientific Technical and Cultural Affairs (IAP no. P7/44), the FRS-FNRS (MIS F.4518.12, IISN 4.4503.11), the Tournesol/Hubert Curien partnership between Belgium and France (R.CFRA.1567), the Agence Nationale de la Recherche (Bactoprenyl project, ANR-11-BSV3-002), the Centre National de la Recherche Scientifique and the University of Paris-Sud (UMR 9198). D.W. was supported, in part, by the Deutsche Forschungsgemeinschaft (German Research Foundation). P.J.S. was supported by grants from the Wellcome Trust and the BBSRC (BB/I019855/1, BB/P01948X/1, BB/R002517/1). MDS were performed using the Irish Centre for High-End Computing (ICHEC) facilities.

## Author contributions

M.E.-G. produced and helped crystallize protein, performed site-directed mutagenesis and functional assays. N.H. crystallized protein, collected synchrotron data and helped prepare the manuscript. C-Y.H., V.O. and M.W. collected and processed synchrotron data, and solved and analyzed the structure. R.W. contributed to data analysis. P.S. performed molecular modeling and MDS. D.W. performed mechanism analysis and advised on the project. T.T., F.K., M.W. and M.C. were responsible for project strategy and management. MC interpreted the structure and wrote the manuscript. Contributions and edits to the written manuscript were provided by co-authors.

## Additional information

**Competing interests:** The authors declare no competing interests.

