## [Peer Review File · Nature Communications]

Reviewers' comments:

Reviewer #1 (Remarks to the Author):

This manuscript describes a crystal structure and biochemical and structure prediction study of BacA, a membrane enzyme involved in the process of synthesizing peptidoglycans, by carrying out pyrophosphatase activity on undecaprenyl-pyrophosphate (55-PP). This remarkable process involves an interesting antibiotic target. This manuscript is a well written, elegantly presented, and the study is thoughtfully designed. It uses a recently developed in meso in situ x-ray crystallography method (IMISX). The structure alone will prove an important foundation for further studies of peptidoglycan synthesis.

The authors solved the structure at high resolution revealing a novel architecture with features only previously observed in coupled membrane transporters. These features include an inverted topology repeat with interdigitation between the repeats, and an asymmetry of the conformations of those 2 repeats, which produces a pathway open to one side of the membrane and creates the active site. Moreover they were able to use this asymmetry, similar to previous studies of transporters, to model an alternate conformation of BacA. Comparing their structure to this model allows conformational mechanism to be proposed and discussed. The authors test this dynamic mechanism using cross-linking experiments and these results suggest that some kind of conformational change is probably involved.

Reviewer #2 (Remarks to the Author):

Overall comments

This paper describes the first crystal structure of BacA, an undecaprenyl-pyrophosphate phosphatase involved in recycling the lipid carrier used in the biosynthesis of important bacterial polysaccharide structures such as peptidoglycan. The 3-dimensional fold of BacA was previously predicted from co-evolution data (Ovchinnikov et al., 2015), and here the authors show that the crystal structure is similar to the predicted fold. The fold consists of two 5-helix repeats related by pseudo rotational symmetry across the membrane, which is often seen in transporters. The cavity of BacA is open to the periplasm, suggesting a periplasmic active site.

The structure of BacA is an important one. However, the manuscript in its current form is not suitable for publication in Nature Communications without major revisions. The manuscript is excessively bloated, and contains too many unsubstantiated claims not to mention the lack of functional studies. The manuscript is focused on speculative "flippase" function of BacA. Having an "interdigitated inverted-topology repeat" is interesting for an enzyme, but not a new discovery (see Ovchinnikov et al., 2015), and claims of its significance (BacA might function as a flippase) are purely speculation.

If authors want to maintain their "flippase" focus of the manuscript, they must demonstrate the flippase function of BacA, which will be a major step forward. Without functional evidence that BacA is a flippase, the manuscript must be revised with the focus of the mechanism of the pyrophosphatase function and its recognition of undecaprenyl group. Authors should show functional studies to substantiate their mechanistic hypotheses by mutating key residues (R174, R261, and E21). Also, recognition of undecaprenyl group by this enzyme would be worthy of discussion but I could not find any discussion regarding that aspect. Authors should compare the recent structures of Und-P recognizing proteins: MraY, MurJ, and ArnT.

Main comments

1. The manuscript should be trimmed significantly (below 4000 words, now ~6000 words),

removing many sections in the introduction and discussion. Especially the flippase part, gene duplication, folding parts in the discussion should be either removed or be very brief.

2. Putative active site (lines 227-265). The authors fail to present evidence for their model of substrate binding. They tried pre-incubation or doping the mesophase with additional substrate, but neither method allowed the substrate to be built unambiguously. Authors should perform mutagenesis experiments on the residues implicated in the binding model or previously untested residues (Arg174, Arg261, His36, Lys46, Lys114 etc) and evaluate their impact in the pyrophosphatase assay. The method section includes the description of the functional assay but I cannot find any functional data presented in the manuscript.

3. Arg174 is shown in proximity to the catalytic Glu21 in Fig. 5. However, the proposed catalytic mechanism requires protonation of Glu21, which would disrupt the salt-bridge with Arg174. The authors also propose Arg174 to stabilize the pyrophosphate, consistent with previously proposed mechanism (Manat et al., 2015), which makes more sense. Please resolve the actual role of Arg174, and illustrate Arg174 closer to the pyrophosphate than to Glu21. In addition, please cite all the essential residues determined by the previous papers (Manat et al., 2015), not just Ser27.

4. Recognition of undecaprenyl group by membrane proteins is increasingly recognized to be important for provide specificity. It would be worthwhile to make a comparison of BacA with the recent structures of undecaprenyl-recognizing membrane proteins: MraY, MurJ, and ArnT. MraY and MurJ are in the same peptidoglycan pathway, which will make the manuscript more interesting as how proteins in the same pathway may recognize Und-P or Und-PP in a similar or different manner.

5. 11a. Proposed catalytic mechanism (lines 266-292, Fig. 5). Calcium is important for activity, possibly by stabilizing the pyrophosphate, and thus should be added to the mechanism figure (see Manat et al. 2015). The authors should consider looking for calcium in their structure by collecting data from crystals at long wavelength and to high redundancy. If their model of calcium binding is correct, they would expect to see an anomalous difference Fourier density peak.

Minor comments

1. Title. The general reader in the bacterial cell wall synthesis would appreciate more functional context, rather than structural detail. Suggest changing to something like "Crystal structure of BacA, an undecaprenyl-pyrophosphatase phosphatase involved in lipid carrier recycling during peptidoglycan biosynthesis."

2. Authors list. One author (R.W.) is listed in the author list but without any author contributions.

3. References: I found that more citations regarding the LCP method, inverted topology, and transporter rather than citing relevant references of BacA and membrane proteins in the bacterial cell wall synthesis. The senior author of the manuscript is undoubtedly the authority in developing the LCP method, but citing > five papers regarding the LCP method is unnecessary. There has been significant progress in the membrane-associated step in the bacterial cell wall synthesis and it would make the paper stronger if it covers what is known in the field.

4. Nomenclature. Please use established nomenclature Und-P/Und-PP or C55-P/C55-PP, instead of 55P/55PP.

5. Figure 2c. The "book" representation of the topology could be misinterpreted to mean mirror symmetry rather than rotational symmetry. Suggest adding two triangles behind the helix cylinders of Fig. 2b to show the relation between the two domains, removing Fig. 2c.

6. Figure 3 is redundant with Fig. 2, and should be combined. Keep Fig. 2a and Fig. 2b (with triangles in the background denoting the inverted topology repeat). Show Fig. 3f next to Fig. 2d. to show the helix numbering and pseudo rotational symmetry. Remove the rest (2c, 3a, 3b, 3c, 3d, 3e).

7. Line 169, "positive inside rule". Suggest adding surface representation colored by electrostatic distribution (e.g. using the APBS plugin in PyMOL) to Extended Data Fig. 4.

8. Figure 4a. Adding a surface representation next to the cartoon representation would be helpful to appreciate the actual size of the cavity. Fig. 4c. has too much superfluous detail (please show the backbone in cartoon, and just the sidechains in sticks).

9. Overall architecture (lines 143-198) and interdigitating inverted-topology repeat fold (lines 199-

226). These two sections should be condensed into one. Description of the fold is too long-winded, repetitive, and difficult to follow. It should be trimmed down substantially.

10. Alternate states (lines 293-317). The authors should tone down their claims of BacA being an alternating access transporter, since there is no evidence beyond the inverted-topology repeats and reminiscence to known transporters such as LeuT. Furthermore, the inward-facing state is purely based on in silico modeling.

11. BacA could be a flippase (Lines 336-387). The authors suggest that BacA might function as a flippase in addition to pyrophosphatase, which is certainly a valid point for discussion. However, they provide too much speculative detail (~800 words!) trying to propose a flippase mechanism. Keep the discussion to the possibility of BacA being a flippase, but remove the speculative mechanisms. Also, comparison with the structure of the lipid flippase MurJ would be helpful to the general readers.

12. Gene duplication (lines 388-416). Gene duplication is common in evolution and obvious from the topology of BacA. Doesn't warrant 400 words and should be removed.

13. BacA with alternate activities (lines 417-430). The section title is confusing. Rename it to "BacA might contain both periplasmic and cytoplasmic active sites". This would be an interesting alternative hypothesis to BacA being a flippase, and would address the long-standing issue of the missing cytoplasmic pyrophosphatase.

14. Possible folding mechanism (lines 431-451). This entire section is too speculative and tangential. This section should be removed.

15. Extended Data Figure 2d. Please show the same topology as the crystal structure (Fig. 2b) but just mention that the prediction failed to define the topology. zExtended Data Figure 6. Would be helpful to show a C-alpha RMSD plot to illustrate regions that differ between the crystal structure and predicted model.

16. Extended Data Figure 8. This figure is redundant with Fig. 2 in the main paper.

17. Extended Data Figure 10. Please explain the role of Arg174.

18. Extended Data Figure 11. Would be helpful to show both cartoon and surface representation.

19. Extended Data. Figure 14. Folding mechanism is too speculative and does not warrant inclusion.

Reviewer #3 (Remarks to the Author):

The manuscript by El Ghachi and co-workers describes the 2.6 Å crystal structure of BacA, a pyrophosphatase shown to play a key role in membrane phospholipid shuttling, and hence cell wall stability and ultimately virulence. The structure was solved thanks to the employment of both Lipidic Cubic Phases and in situ X-ray crystallographic approaches. These technologies allowed the collection of data from hundreds of micro crystals without the need for removing them from their site of growth. The structure reveals an interdigitated inverted topology repeat, which has been observed in transporters and channels (activities not originally proposed for BacA). The morphing sequences between open/closed forms of the protein, followed by validation through the generation of disulfides, are particularly useful in understanding BacA activity. This particularly well-written and insightful manuscript should be acceptable for publication in Nature Communications once a few minor points have been addressed.

Line 77: different proteins have been suggested as acting as Lipid II flippases, not only MurJ. These proteins should be mentioned and the works referenced

Lines 212-226: this description is rather long and could be shortened, considering the clarity of the figures. The same can be said regarding the gene duplication discussion (lines 388-416).

Lines 296-305: considering that BacA can exist in inward- and outward-facing states, it is conceivable that authors could have crystallized the protein in both conformations, but this is not commented upon. Were both forms crystallized, but only one diffracted well? Or was the

'periplasm-facing' form, described here, favored? If so, it would be interesting to speculate as to why a cytoplasmic-facing form could not be crystallized and how such a form could be stabilized.

Methods: authors should comment on the sites of Hg binding, and if this could have somehow influenced the inter-helical interactions. Are there reasons as to why a less time-consuming approach (i.e. phasing with SeMet) was not employed?

Extended Figure 3: it would be of interest for readers if zoomed images could be included both for protein-lipid and protein-protein contacts

It seems to this reviewer that the region between alpha3-and alpha4- displays the longest loop region, and a quick sequence alignment suggests that this region could display the greatest differences among BacA molecules from different bacteria. Can authors speculate on the function of this region? Could BacA be interacting with other PG-forming molecules and if so have partners been identified?

Reviewers' comments:

Reviewer #1 (Remarks to the Author):

This manuscript describes a crystal structure and biochemical and structure prediction study of BacA, a membrane enzyme involved in the process of synthesizing peptidoglycans, by carrying out pyrophosphatase activity on undecaprenyl-pyrophosphate (55-PP). This remarkable process involves an interesting antibiotic target. This manuscript is a well written, elegantly presented, and the study is thoughtfully designed. It uses a recently developed in meso in situ x-ray crystallography method (IMISX). The structure alone will prove an important foundation for further studies of peptidoglycan synthesis.

The authors solved the structure at high resolution revealing a novel architecture with features only previously observed in coupled membrane transporters. These features include an inverted topology repeat with interdigitation between the repeats, and an asymmetry of the conformations of those 2 repeats, which produces a pathway open to one side of the membrane and creates the active site. Moreover they were able to use this asymmetry, similar to previous studies of transporters, to model an alternate conformation of BacA. Comparing their structure to this model allows conformational mechanism to be proposed and discussed. The authors test this dynamic mechanism using cross-linking experiments and these results suggest that some kind of conformational change is probably involved.

**** We thank the Reviewer for the complimentary remarks.**

Reviewer #2 (Remarks to the Author):

Overall comments

This paper describes the first crystal structure of BacA, an undecaprenyl-pyrophosphate phosphatase involved in recycling the lipid carrier used in the biosynthesis of important bacterial polysaccharide structures such as peptidoglycan. The 3-dimensional fold of BacA was previously predicted from co-evolution data (Ovchinnikov et al., 2015), and here the authors show that the crystal structure is similar to the predicted fold. The fold consists of two 5-helix repeats related by pseudo rotational symmetry across the membrane, which is often seen in transporters. The cavity of BacA is open to the periplasm, suggesting a periplasmic active site.

The structure of BacA is an important one. However, the manuscript in its current form is not suitable for publication in Nature Communications without major revisions. The manuscript is excessively bloated, and contains too many unsubstantiated claims not to mention the lack of functional studies. The manuscript is focused on speculative “flippase” function of BacA. Having an “interdigitated inverted-topology repeat” is interesting for an enzyme, but not a new discovery (see Ovchinnikov et al., 2015), and claims of its significance (BacA might function as a flippase) are purely speculation.

If authors want to maintain their “flippase” focus of the manuscript, they must demonstrate the flippase function of BacA, which will be a major step forward. Without functional evidence that BacA is a flippase, the manuscript must be revised with the focus of the mechanism of the pyrophosphatase function and its recognition of undecaprenyl group. Authors should show functional studies to substantiate their mechanistic hypotheses by mutating key residues (R174, R261, and E21). Also, recognition of undecaprenyl group by this enzyme would be worthy of discussion but I could not find any discussion regarding that aspect. Authors should compare the recent structures of Und-P recognizing proteins: MraY, MurJ, and ArnT.

**** We thank the Reviewer for the time, effort and care taken to provide this detailed and valuable critique.**

Main comments

1. The manuscript should be trimmed significantly (below 4000 words, now ~6000 words), removing many sections in the introduction and discussion. Especially the flippase part, gene duplication, folding parts in the discussion should be either removed or be very brief.

**** Following the recommendations of the Reviewer, the text has been trimmed considerably.**

2. Putative active site (lines 227-265). The authors fail to present evidence for their model of substrate binding. They tried pre-incubation or doping the mesophase with additional substrate, but neither method allowed the substrate to be built unambiguously. Authors should perform mutagenesis experiments on the residues implicated in the binding model or previously untested residues (Arg174, Arg261, His36, Lys46, Lys114 etc) and evaluate their impact in the pyrophosphatase assay.

**** Extensive mutagenesis studies of how select residues impact on the pyrophosphatase activity of BacA have been performed and reported on in the literature. Critical residues identified in those studies that include Glu17, Glu21, Ser27, Ser173, Arg174, Thr178 and Arg261 are now listed in Extended Data Table 2. Some of the mutagenesis work was done by authors of the current manuscript. His36, Lys46 and Lys114, referred to by the Reviewer, are on the periplasmic rim of the binding pocket and were proposed to facilitate phosphate exit following catalysis. A role in catalysis is not expected.**

The most convincing way to identify residues that are directly involved in substrate binding and catalysis, as distinct from conformational stability, is to obtain structures of complexes between the native or inactive mutant forms of the enzyme and its substrate, products and intermediates. However, this is a major project in itself and beyond the scope of the current work.

The method section includes the description of the functional assay but I cannot find any functional data presented in the manuscript.

**** Activity measurements were/are reported under 'Structural changes during catalysis' and in Extended Data Table 2.**

3. Arg174 is shown in proximity to the catalytic Glu21 in Fig. 5. However, the proposed catalytic mechanism requires protonation of Glu21, which would disrupt the salt-bridge with Arg174. The authors also propose Arg174 to stabilize the pyrophosphate, consistent with previously proposed mechanism (Manat et al., 2015), which makes more sense. Please resolve the actual role of Arg174, and illustrate Arg174 closer to the pyrophosphate than to Glu21.

**** The proposed mechanism (current Fig. 4) has been modified slightly, as suggested by the Reviewer. A possible role for Arg174 in stabilizing the pyrophosphate and pentavalent intermediates is referred to in the revised version of Fig. 4.**

In addition, please cite all the essential residues determined by the previous papers (Manat et al., 2015), not just Ser27.

**** Critical residues in BacA have been identified by Manat et al., (2015) and by Chang et al., (2014) on the basis of extensive mutagenesis studies. These are listed in Extended Data Table 2. Both studies show Arg174 to serve a key role in substrate binding and/or catalysis, consistent with the current study.**

4. Recognition of undecaprenyl group by membrane proteins is increasingly recognized to be important for provide specificity. It would be worthwhile to make a comparison of BacA with the recent structures of undecaprenyl-recognizing membrane proteins: MraY, MurJ, and ArnT. MraY and MurJ are in the same peptidoglycan pathway, which will make the manuscript more interesting as how proteins in the same pathway may recognize Und-P or Und-PP in a similar or different manner.

**** As noted in the manuscript, we have attempted to obtain a structure of BacA in complex with its lipid substrate and product but have failed to identify convincing electron density for either. At this point therefore, placement of the undecaprenyl group in the BacA structure would be entirely speculative. Relatedly, our MDS analysis (BacA-Extended Data Movie 1A) shows what we consider to be a physico-chemically reasonable location for this long, multiply branched, highly disordered polyisoprenyl chain extending away from the protein into the apolar recesses of the membrane.**

A conserved recognition/binding site for the isoprenyl moiety is not obvious in any of the three structures referred to by the Reviewer. Indeed, the MraY and MurJ models have nothing concrete to contribute regarding undecaprenyl group placement. In the ArnT study, the electron density ascribed to a polyisoprenyl moiety is fragmented. Further, given that the reported resolution in the ArnT study was 3.2 Å and the structure was obtained by the LCP method in a mesophase that is 2 molar monoolein, one must be careful when ascribing unaccounted-for density to anything other than 'adventitious' monoolein.

5. 11a. Proposed catalytic mechanism (lines 266-292, Fig. 5). Calcium is important for activity, possibly by stabilizing the pyrophosphate, and thus should be added to the mechanism figure (see Manat et al. 2015). The authors should consider looking for calcium in their structure by collecting data from crystals at long wavelength and to high redundancy. If their model of calcium binding is correct, they would expect to see an anomalous difference Fourier density peak.

**** Indeed, calcium was looked for as part of this study. Measurements were made using 1.9 Å (6.526 keV) X-rays providing a resolution of 2.6 Å and a multiplicity of 18. However, the only peaks in the anomalous difference map were at the two mercury sites. None was observed for calcium or sulfur. This result does not rule out a role for calcium in the reaction. Calcium may well be there in the binding pocket at low occupancy and/or disordered. It may enter along with the substrate.**

Despite not having direct structural evidence in support of calcium's presence in the binding pocket of BacA, the revised version of the legend to the reaction mechanism figure (Fig. 4) now makes reference to calcium, as recommended by the Reviewer. Separately, calcium has been shown to considerably enhance BacA's pyrophosphorylase activity in *E. coli* (Cheng et al., 2014; Manat et al., 2015).

Minor comments

1. Title. The general reader in the bacterial cell wall synthesis would appreciate more functional context, rather than structural detail. Suggest changing to something like "Crystal structure of BacA, an undecaprenyl-pyrophosphatase phosphatase involved in lipid carrier recycling during peptidoglycan biosynthesis."

**** The title has been changed to "Crystal structure of the undecaprenyl-pyrophosphate phosphatase, BacA, and its role in peptidoglycan biosynthesis."**

2. Authors list. One author (R.W.) is listed in the author list but without any author contributions.

**** We thank the Reviewer for alerting us to the omission. It has been corrected.**

3. References: I found that more citations regarding the LCP method, inverted topology, and transporter rather than citing relevant references of BacA and membrane proteins in the bacterial cell wall synthesis. The senior author of the manuscript is undoubtedly the authority in developing the LCP method, but citing > five papers regarding the LCP method is unnecessary. There has been significant progress in the membrane-associated step in the bacterial cell wall synthesis and it would make the paper stronger if it covers what is known in the field.

**** Agreed. Non-essential references to the LCP method have been removed and more citations concerning bacterial cell wall synthesis have been added.**

4. Nomenclature. Please use established nomenclature Und-P/Und-PP or C55-P/C55-PP, instead of 55P/55PP.

**** We have adopted the C55P/C55PP nomenclature.**

5. Figure 2c. The “book” representation of the topology could be misinterpreted to mean mirror symmetry rather than rotational symmetry. Suggest adding two triangles behind the helix cylinders of Fig. 2b to show the relation between the two domains, removing Fig. 2c.

****BacA consists of two interdigitated inverted-topology repeats. The interdigitation and the creation of domains from parts of the two repeats make for a complicated topology that can be difficult to comprehend. We have found that using the open/closed book to explain how the two domains form as the repeats interleave is relatively easily understood. It is for this reason that we prefer to retain Fig. 2c.**

6. Figure 3 is redundant with Fig. 2, and should be combined. Keep Fig. 2a and Fig. 2b (with triangles in the background denoting the inverted topology repeat). Show Fig. 3f next to Fig. 2d. to show the helix numbering and pseudo rotational symmetry. Remove the rest (2c, 3a, 3b, 3c, 3d, 3e).

**** Much thought went into composing Fig. 2 and respectfully we prefer to keep it as is. To eliminate redundancy in the main section of the manuscript, Fig. 3 has been moved to the Extended Data section (Extended Data Fig. 4). It is important to retain the several views of BacA in this figure in light of the complex topology of the enzyme.**

7. Line 169, “positive inside rule”. Suggest adding surface representation colored by electrostatic distribution (e.g. using the APBS plugin in PyMOL) to Extended Data Fig. 4.

**** A surface representation where cationic and anionic residues are highlighted has been included in Extended Data Fig. 5.**

8. Figure 4a. Adding a surface representation next to the cartoon representation would be helpful to appreciate the actual size of the cavity.

**** Surface representation figures have been included (Fig. 3b)**

Fig. 4c. has too much superfluous detail (please show the backbone in cartoon, and just the sidechains in sticks).

**** We agree with the Reviewer that Fig. 3e (originally Fig. 4c) includes a lot of detail. However, the detail in the figure which relates to how the polar end of the substrate likely interacts with the binding pocket matches exactly what is in the legend. We feel it is important to keep this figure.**

9. Overall architecture (lines 143-198) and interdigitating inverted-topology repeat fold (lines 199-226). These two sections should be condensed into one. Description of the fold is too long-winded, repetitive, and difficult to follow. It should be trimmed down substantially.

**** The two sections have been combined and shortened. However, we are reluctant to condense these important descriptive sections to such an extent that the complex nature of the topology in BacA is not fully explained.**

10. Alternate states (lines 293-317). The authors should tone down their claims of BacA being an alternating access transporter, since there is no evidence beyond the inverted-topology repeats and reminiscence to known transporters such as LeuT. Furthermore, the inward-facing state is purely based on in silico modeling.

**** Nowhere in the manuscript do we claim that BacA operates as an alternating access transporter. Rather, the fact that BacA has an IITR fold reminiscent of certain channels and transporters**

suggested that switching between inward- and outward-facing states might be possible by asymmetry exchange. This was indeed observed to happen and is described as an interesting feature of this extraordinary small membrane protein.

11. BacA could be a flippase (Lines 336-387). The authors suggest that BacA might function as a flippase in addition to pyrophosphatase, which is certainly a valid point for discussion. However, they provide too much speculative detail (~800 words!) trying to propose a flippase mechanism. Keep the discussion to the possibility of BacA being a flippase, but remove the speculative mechanisms. Also, comparison with the structure of the lipid flippase MurJ would be to the general readers.

**** This section has been shortened. MurJ is a member of the mouse virulence factor family of the MOP transporter superfamily that includes the MATE and ORF families. It has 14 transmembrane helices and is proposed to function as a lipid II flippase by a rocker-switch mechanism. In many ways, MurJ is a very different protein to BacA and its proposed mechanism of action distinct from that for BacA. We feel that a comparison of MurJ and BacA is not warranted.**

12. Gene duplication (lines 388-416). Gene duplication is common in evolution and obvious from the topology of BacA. Doesn't warrant 400 words and should be removed.

**** The section dealing with gene duplication is important for context. It has been reduced in size considerably.**

13. BacA with alternate activities (lines 417-430). The section title is confusing. Rename it to "BacA might contain both periplasmic and cytoplasmic active sites". This would be an interesting alternative hypothesis to BacA being a flippase, and would address the long-standing issue of the missing cytoplasmic pyrophosphatase.

**** For clarity, the section title has been revised to "BacA with alternating active sites on either side of the membrane?"**

14. Possible folding mechanism (lines 431-451). This entire section is too speculative and tangential. This section should be removed.

**** Given the complex topology adopted by BacA it is intriguing to consider how folding gives rise to this extraordinary interdigitated inverted-topology repeat fold. We agree that what we propose is speculative but that is what a discussion section is for. Furthermore, we feel that the proposal is entirely reasonable, it makes sense. As such, it amounts to a testable hypothesis that we would prefer to retain.**

15. Extended Data Figure 2d. Please show the same topology as the crystal structure (Fig. 2b) but just mention that the prediction failed to define the topology. zExtended Data Figure 6.

**** Done.**

Would be helpful to show a C-alpha RMSD plot to illustrate regions that differ between the crystal structure and predicted model.

**** Included in Extended Data Fig. 7.**

16. Extended Data Figure 8. This figure is redundant with Fig. 2 in the main paper.

**** This figure has been removed.**

17. Extended Data Figure 10. Please explain the role of Arg174.

**** Done.**

18. Extended Data Figure 11. Would be helpful to show both cartoon and surface representation.

**** Surface representations have been added to Fig. 3.**

19. Extended Data. Figure 14. Folding mechanism is too speculative and does not warrant inclusion.

**** See response to Item 14 above.**

Reviewer #3 (Remarks to the Author):

The manuscript by El Ghachi and co-workers describes the 2.6 Å crystal structure of BacA, a pyrophosphatase shown to play a key role in membrane phospholipid shuttling, and hence cell wall stability and ultimately virulence. The structure was solved thanks to the employment of both Lipidic Cubic Phases and in situ X-ray crystallographic approaches. These technologies allowed the collection of data from hundreds of micro crystals without the need for removing them from their site of growth. The structure reveals an interdigitated inverted topology repeat, which has been observed in transporters and channels (activities not originally proposed for BacA). The morphing sequences between open/closed forms of the protein, followed by validation through the generation of disulfides, are particularly useful in understanding BacA activity. This particularly well-written and insightful manuscript should be acceptable for publication in Nature Communications once a few minor points have been addressed.

**** We thank the Reviewer for the complimentary remarks.**

Line 77: different proteins have been suggested as acting as Lipid II flippases, not only MurJ. These proteins should be mentioned and the works referenced

**** Reference to RodA and FtsW as possible flippases has been included in the revised legend to Fig. 1.**

Insert/massage into legend of Fig. 1 with references. Occur. ...RodA and FtsW may also be involved in flipping lipid II. 55PP..... reference Meeske et al., 2016 Nature doi:10.1038/nature19331 for RodA and Mohammadi et al., EMBO Journal (2011) 30, 1425–1432 for FtsW.

Lines 212-226: this description is rather long and could be shortened, considering the clarity of the figures.

**** This paragraph has been shortened.**

The same can be said regarding the gene duplication discussion (lines 388-416).

**** This section has been shortened.**

Lines 296-305: considering that BacA can exist in inward- and outward-facing states, it is conceivable that authors could have crystallized the protein in both conformations, but this is not commented upon. Were both forms crystallized, but only one diffracted well? Or was the 'periplasm-facing' form, described here, favored? If so, it would be interesting to speculate as to why a cytoplasmic-facing form could not be crystallized and how such a form could be stabilized.

**** It is entirely possible that what the Reviewer suggests actually happened: We crystallized the alternate state but because of low resolution it was not possible to tell the form and we never followed up on it. With a view to capturing the inward-facing state, we propose to crystallize the double Cys mutant used for cross-linking where the functional data suggest that the protein has been locked in the 'cytoplasm-facing' state. This represents an extension of the current study and is planned.**

Methods: authors should comment on the sites of Hg binding, and if this could have somehow influenced

the inter-helical interactions. Are there reasons as to why a less time-consuming approach (i.e. phasing with SeMet) was not employed?

**** SAD phasing with SeMet was attempted. However, the labelled protein was unstable and was not suitable for crystallization. In an early, longer version of the manuscript we included data on crystals grown in the presence of the lipid product C55P (but without mercury) that provided a structure to 2.25 Å resolution. However, because convincing density for the lipid was not observed and in the interests of space, the data were removed. We note that the BacA-Hg and 'BacA-C55P' structures were essentially identical ruling out any effect of mercury on inter-helical interactions.**

Extended Figure 3: it would be of interest for readers if zoomed images could be included both for protein-lipid and protein-protein contacts

**** Extended Data Fig. 3 has been modified and enlarged to show both interaction types.**

It seems to this reviewer that the region between alpha3-and alpha4- displays the longest loop region, and a quick sequence alignment suggests that this region could display the greatest differences among BacA molecules from different bacteria. Can authors speculate on the function of this region? Could BacA be interacting with other PG-forming molecules and if so have partners been identified?

**** The loop in question is on the cytoplasmic side of the protein and is 27 residues long. With 2 anionic and 7 cationic residues it bears a net positive charge. This might contribute to steering negatively charged C55P and/or C55PP toward or away from the enzyme should the cytoplasmic side of BacA bear an active or binding site, as discussed. We are not aware of any data in the literature indicating that BacA interacts directly with other partners.**

Reviewers' Comments:

Reviewer #2:

Remarks to the Author:

The authors revised their manuscript according to suggestions by changing the title and toning down their flippase claim. The structure of this enzyme is an important contribution to the field. I have no further comments.

Reviewer #3:

Remarks to the Author:

The authors have replied to all of my queries and the manuscript is now acceptable for publication.